# THE BURES METRIC FOR TAMING MODE COLLAPSE IN GENERATIVE ADVERSARIAL NETWORKS

## ABSTRACT

Generative Adversarial Networks (GANs) are performant generative methods yielding high-quality samples. However, under certain circumstances, the training of GANs can lead to mode collapse or mode dropping, i.e. the generative models not being able to sample from the entire probability distribution. To address this problem, we use the last layer of the discriminator as a feature map to study the distribution of the real and the fake data. During training, we propose to match the real batch diversity to the fake batch diversity by using the Bures distance between covariance matrices in feature space. The computation of the Bures distance can be conveniently done in either feature space or kernel space in terms of the covariance and kernel matrix respectively. We observe that diversity matching reduces mode collapse substantially and has a positive effect on the sample quality. On the practical side, a very simple training procedure, that does not require additional hyperparameter tuning, is proposed and assessed on several datasets.

## 1 INTRODUCTION

In several machine learning applications, data is assumed to be sampled from an implicit probability distribution. The estimation of this empirical implicit distribution is often intractable, especially in high dimensions. To tackle this issue, generative models are trained to provide an algorithmic procedure for sampling from this unknown distribution. Popular approaches are Variational Auto-Encoders proposed by Kingma & Welling (2014), Generating Flow models by Rezende & Mohamed (2015) and Generative Adversarial Networks (GANs) initially developed by Goodfellow et al. (2014). The latter are particularly successful approaches to produce high quality samples, especially in the case of natural images, though their training is notoriously difficult. The vanilla GAN consists of two networks: a generator and a discriminator. The generator maps random noise, usually drawn from a multivariate normal, to fake data in input space. The discriminator estimates the likelihood ratio of the generator network to the data distribution. It often happens that a GAN generates samples only from a few of the many modes of the distribution. This phenomenon is called 'mode collapse'.

**Contribution.** We propose BuresGAN: a generative adversarial network which has the objective function of a vanilla GAN complemented by an additional term, which is given by the squared Bures distance between the covariance matrix of real and fake batches in a latent space. This loss function promotes a matching of fake and real data in a feature space $\mathbb{R}^f$, so that mode collapse is reduced. Conveniently, the Bures distance also admits both a feature space and kernel based expression. Contrary to other related approaches such as in Che et al. (2017) or Srivastava et al. (2017), the architecture of the GAN is unchanged, only the objective is modified. A variant called alt-BuresGAN, which is trained with alternating minimization, achieves competitive performance with a simple training procedure that does not require hyperparameter tuning or an additional regularization such as a gradient penalty. We empirically show that the proposed methods are robust when it comes to the choice of architecture and do not require an additional fine architecture search. Finally, an extra asset of BuresGAN is that it yields competitive or improved IS and FID scores compared with the state of the art on CIFAR-10 and STL-10 using a ResNet architecture.

**Related works.** The Bures distance is closely related to the Fréchet distance (Dowson & Landau, 1982) which is a 2-Wasserstein distance between multivariate normal distributions. Namely,

the Fréchet distance between multivariate normals of equal means is the Bures distance between their covariance matrices. The Bures distance is also equivalent to the exact expression for the 2-Wasserstein distance between two elliptically contoured distributions with the same mean as shown in Gelbrich (1990) and Peyré et al. (2019). Noticeably, the Fréchet Inception Distance score (FID) is a popular manner to assess the quality of generative models. This score uses the Fréchet distance between real and generated samples in the feature space of a pre-trained inception network as it is explained in Salimans et al. (2016) and Heusel et al. (2017).

There exist numerous works aiming to improve training efficiency of generative networks. For mode collapse evaluation, we compare BuresGAN to the most closely related works. GDPP-GAN (Elfeki et al., 2019) and VEEGAN (Srivastava et al., 2017) also try to enforce diversity in 'latent' space. GDPP-GAN matches the eigenvectors and eigenvalues of the real and fake diversity kernel. In VEE-GAN, an additional reconstructor network is introduced to map the true data distribution to Gaussian random noise. In a similar way, architectures with two discriminators are analysed by Nguyen et al. (2017), while MADGAN (Ghosh et al., 2018) uses multiple discriminators and generators. A different approach is taken by Unrolled-GAN (Metz et al., 2017) which updates the generator with respect to the unrolled optimization of the discriminator. This allows the training to be adjusted between using the optimal discriminator in the generator's objective, which is ideal but infeasible in practice. Wasserstein GANs (Arjovsky et al., 2017; Gulrajani et al., 2017) leverage the 1-Wasserstein distance to match the real and generated data distributions. In MDGAN (Che et al., 2017), a regularization is added to the objective function, so that the generator can take advantage of another similarity metric with more predictable behavior. This idea is combined with a penalization of the missing modes. More recent approaches to reducing mode collapse are variations of WGAN (Wu et al., 2018). Entropic regularization has been also proposed in PresGAN (Dieng et al., 2020), while metric embeddings were used in the paper introducing BourGAN (Xiao et al., 2018). A simple packing procedure which significantly reduces mode collapse was proposed in PacGAN (Lin et al., 2018) that we also consider hereafter in our comparisons.

## 2 METHOD

A GAN consists of a discriminator $D : \mathbb{R}^d \to \mathbb{R}$ and a generator $G : \mathbb{R}^\ell \to \mathbb{R}^d$ which are typically defined by neural networks and parametrized by real vectors. The value $D(\boldsymbol{x})$ gives the probability that $\boldsymbol{x}$ comes from the empirical distribution, while the generator $G$ maps a point $\boldsymbol{z}$ in the latent space $\mathbb{R}^\ell$ to a point in the input space $\mathbb{R}^d$. The training of a GAN consists in solving

$$\min_G \max_D \mathbb{E}_{\boldsymbol{x} \sim p_d}[\log D(\boldsymbol{x})] + \mathbb{E}_{\boldsymbol{x} \sim p_g}[\log(1 - D(\boldsymbol{x}))], \tag{1}$$

by alternating two phases of training. In equation 1, the expectation in the first term is over the empirical data distribution $p_d$, while the expectation in the second term is over the generated data distribution $p_g$, implicitly given by the mapping by $G$ of the latent prior distribution $\mathcal{N}(0, \mathbb{I}_\ell)$. It is common to define and minimize the discriminator loss by

$$V_D = -\mathbb{E}_{\boldsymbol{x} \sim p_d}[\log D(\boldsymbol{x})] - \mathbb{E}_{\boldsymbol{x} \sim p_g}[\log(1 - D(\boldsymbol{x}))]. \tag{2}$$

In practice, it is proposed in Goodfellow et al. (2014) to minimize generator loss

$$V_G = -\mathbb{E}_{\boldsymbol{z} \sim \mathcal{N}(0, \mathbb{I}_\ell)}[\log D(G(\boldsymbol{z}))], \tag{3}$$

rather than the second term of equation 1, for an improved training efficiency.

**Matching real and fake data covariance.** To prevent mode collapse, we encourage the generator to sample fake data of similar diversity to the real data. This is achieved by matching the sample covariance matrices of the real and fake data respectively. Covariance matching and similar ideas were explored for GANs in Mroueh et al. (2017) and Elfeki et al. (2019). In order to compare covariance matrices, we propose to use the squared Bures distance between positive semi-definite $\ell \times \ell$ matrices (Bhatia et al., 2019), i.e.,

$$\mathcal{B}(A, B)^2 = \min_{U \in O(\ell)} \|A^{1/2} - B^{1/2}U\|_F^2 = \text{Tr}(A + B - 2(A^{\frac{1}{2}}BA^{\frac{1}{2}})^{\frac{1}{2}}).$$

Being a Riemannian metric on the manifold of positive semi-definite matrices (Massart & Absil, 2020), the Bures metric is adequate to compare covariance matrices. The covariances are

defined in a feature space associated to the discriminator. More precisely, the last layer of the discriminator, denoted by $\phi(\boldsymbol{x}) \in \mathbb{R}^f$, defines a feature map, namely $D(\boldsymbol{x}) = \sigma(\boldsymbol{w}^\top \phi(\boldsymbol{x}))$, where $\boldsymbol{w}$ is the weight vector of the last dense layer and $\sigma$ is the sigmoid function. We use the normalization $\bar{\phi}(\boldsymbol{x}) = \phi(\boldsymbol{x})/\|\phi(\boldsymbol{x})\|_2$, after the centering of $\phi(\boldsymbol{x})$. Then, we define a covariance matrix as follows: $C(p) = \mathbb{E}_{\boldsymbol{x} \sim p}[\bar{\phi}(\boldsymbol{x})\bar{\phi}(\boldsymbol{x})^\top]$. For simplicity, we denote the real data and generated data covariance matrices by $C_d = C(p_d)$ and $C_g = C(p_g)$, respectively. Our proposal is to replace the generator loss by $V_G + \lambda \mathcal{B}(C_d, C_g)^2$. The value $\lambda = 1$ was found to yield good results in the studied datasets. Two specific training algorithms are proposed. Algorithm 1 deals with the squared Bures distance as an additive term to the generator loss, while an alternating training is used in Algorithm 2 and does not introduce an extra parameter.

| **Algorithm 1:** BuresGAN | **Algorithm 2:** Alt-BuresGAN |
|---|---|
| Sample a real and fake batch ; | Sample a real and fake batch ; |
| Update $G$ by minimizing $V_G + \lambda \mathcal{B}(\hat{\boldsymbol{C}}_r, \hat{\boldsymbol{C}}_g)^2$; | Update $G$ by minimizing $\mathcal{B}(\hat{\boldsymbol{C}}_r, \hat{\boldsymbol{C}}_g)^2$; |
| | Update $G$ by minimizing $V_G$; |
| Update $D$ by maximizing $-V_D$; | Update $D$ by maximizing $-V_D$; |

The training described in Algorithm 1 is analogous to the training of GDPP GAN, although the additional generator loss is rather different. The computational advantage of the Bures distance is that it admits two expressions which can be evaluated numerically in a stable way. Namely, there is no need to calculate a gradient update through an eigendecomposition.

**Feature space expression.** In the training procedure, real $\boldsymbol{x}_i^{(d)}$ and fake data $\boldsymbol{x}_i^{(g)}$ with $i = 1, \ldots, b$ are sampled respectively from the empirical distribution and the mapping of the normal distribution $\mathcal{N}(0, \mathbb{I}_\ell)$ by the generator. Consider the case where the batch size $b$ is larger than the feature space dimension. Let the embedding of the batches in feature space be $\Phi_\alpha = [\phi(\boldsymbol{x}_1^{(\alpha)}), \ldots, \phi(\boldsymbol{x}_b^{(\alpha)})]^\top \in \mathbb{R}^{b \times f}$ with $\alpha = d, g$. The covariance matrix of one batch in feature space[1] is $\hat{C} = \bar{\Phi}^\top \bar{\Phi}$, where $\bar{\Phi}$ is the $\ell_2$-normalized centered feature map of the batch. Numerical instabilities can be avoided by adding a small number, e.g. $1e-14$, to the diagonal elements of the covariance matrices, so that, in practice, we only deal with strictly positive definite matrices. From the computational perspective, an interesting alternative expression for the Bures distance is given by

$$\mathcal{B}(C_d, C_g)^2 = \mathrm{Tr}\big(C_d + C_g - 2(C_g C_d)^{\frac{1}{2}}\big), \tag{4}$$

whose computation requires only one matrix square root. This identity can be obtained from Lemma 1. Note that an analogous result is proved in Oh et al. (2020).

**Lemma 1.** *Let $A$ and $B$ be $f \times f$ symmetric positive semidefinite matrices and let $B = Y^\top Y$. Then, we have: (i) $AB$ is diagonalizable with nonnegative eigenvalues, and (ii) $\mathrm{Tr}((AB)^{\frac{1}{2}}) = \mathrm{Tr}((YAY^\top)^{\frac{1}{2}})$.*

**Kernel based expression.** Alternatively, if the feature space dimension $f$ is larger than the batch size $b$, it is more efficient to compute $\mathcal{B}(\hat{C}_d, \hat{C}_g)$ thanks to $b \times b$ kernel matrices: $K_d = \bar{\Phi}_d \bar{\Phi}_d^\top$, $K_g = \bar{\Phi}_g \bar{\Phi}_g^\top$ and $K_{dg} = \bar{\Phi}_d \bar{\Phi}_g^\top$. Then, we have the kernel based expression

$$\mathcal{B}(\hat{C}_d, \hat{C}_g)^2 = \mathrm{Tr}\left(K_d + K_g - 2\left(K_{dg} K_{dg}^\top\right)^{\frac{1}{2}}\right), \tag{5}$$

which allows to calculate the Bures distance between covariance matrices by computing a matrix square root of a $b \times b$ matrix. This is a consequence of Lemma 2.

**Lemma 2.** *The matrices $X^\top X Y^\top Y$ and $Y X^\top X Y^\top$ are diagonalizable with nonnegative eigenvalues and share the same non-zero eigenvalues.*

**Connection with Wasserstein GAN and integral probability metrics.** The Bures distance is proportional to the 2-Wasserstein distance $\mathcal{W}_2$ between two ellipically contoured distributions, with the same mean (Gelbrich, 1990). For instance, in the case of multivariate normal distributions, we have

$$\mathcal{B}(A, B)^2 = \min_\pi \mathbb{E}_{(X,Y) \sim \pi} \|X - Y\|_2^2 \text{ s.t. } X \sim \mathcal{N}(0, A) \text{ and } Y \sim \mathcal{N}(0, B),$$

---

[1] For simplicity, we omit the normalization by $\frac{1}{b-1}$ in front of the covariance matrix.

where the minimization is over the joint distributions $\pi$. More precisely, in this paper, we make the approximation that the implicit distribution of the real and generated data in the feature space $\mathbb{R}^f$ (associated to $\phi(\boldsymbol{x})$) are elliptically contoured with the same mean. Under different assumptions, the Generative Moment Matching Networks (Ren et al., 2016; Li et al., 2017) work in the same spirit, but use a different approach to match covariance matrices. On the contrary, WGAN uses the Kantorovich dual formula for the 1-Wasserstein distance: $\mathcal{W}_1(\alpha, \beta) = \sup_{f \in \mathrm{Lip}} \int f \, \mathrm{d}(\alpha - \beta)$, where $\alpha, \beta$ are signed measures. Generalizations of such integral formulae are called integral probability metrics (see for instance Binkowski et al. (2018)). Here, $f$ is the discriminator, so that the maximization over Lipschitz functions $f$ plays the role of the maximization over discriminator parameters in the min-max game of equation 1. Then, in the training procedure, this maximization alternates with a minimization over the generator parameters.

We can now discuss the connection with Wasserstein GAN. Coming back to the definition of Bures-GAN, we can now explain that the 2-Wasserstein distance provides an upper bound on an integral probability metric. Then, if we assume that the densities are elliptically contoured distributions in feature space, the use of the Bures distance to calculate $\mathcal{W}_2$ allows to spare the maximization over the discriminator parameters – and this motivates why the optimization of $\mathcal{B}$ only influences updates of the generator in Algorithm 1 and Algorithm 2. Going more into detail, the 2-Wasserstein distance between two probability densities (w.r.t. the same measure) is equivalent to a Sobolev dual norm, which can be interpreted as an integral probability metric. Indeed, let the Sobolev semi-norm $\|f\|_{H^1} = (\int \|\nabla f(x)\|^2 \, \mathrm{d}x)^{1/2}$. Then, its dual norm over signed measures is defined as $\|\nu\|_{H^{-1}} = \sup_{\|f\|_{H^1} \leq 1} \int f \, \mathrm{d}\nu$. It is then shown in Peyre (2018) and Peyré et al. (2019) that there exist two positive constants $c_1$ and $c_2$ such that

$$c_1 \|\alpha - \beta\|_{H^{-1}} \leq \mathcal{W}_2(\alpha, \beta) \leq c_2 \|\alpha - \beta\|_{H^{-1}}.$$

Hence, the 2-Wasserstein distance gives an upper bound on an integral probability metric.

**Algorithmic details.** The matrix square root in equation 4 and equation 5 is obtained thanks to the Newton-Schultz algorithm which is inversion free and can be efficiently calculated on GPUs since it involves only matrix products. In practice, we found 15 iterations of this algorithm to be sufficient for the small scale datasets, while 20 iterations were used for the ResNet examples. A small regularization term $1e-14$ is added for stability. The latent prior distribution is $\mathcal{N}(0, \mathbb{I}_\ell)$ with $\ell = 100$ and the parameter in Algorithm 1 is always set to $\lambda = 1$. In the tables hereafter, we indicate the largest scores in bold, although we invite the reader to also consider the standard deviation.

## 3 EMPIRICAL EVALUATION OF MODE COLLAPSE

The BuresGAN and alt-BuresGAN performances on synthetic data, artificial and real images are compared with the standard DCGAN (Salimans et al., 2016), WGAN-GP, MDGAN, Unrolled GAN, VEEGAN, GDPP and PacGAN. We want to emphasize that the purpose of this experiment is not to challenge these baselines, but to report the improvement obtained by adding the Bures metric to the objective function. It would be straightforward to add the Bures loss to other GAN variants, as well as most GAN architectures, and we would expect an improvement in mode coverage and generation quality. In the experiments, we notice that adding the Bures loss to the vanilla GAN already significantly improves the results.

A low dimensional feature space ($f = 128$) is used for the synthetic data so that the feature space formula in equation 4 is used, while the dual formula in equation 5 is used for the image datasets (Stacked MNIST, CIFAR-10, CIFAR-100 and STL-10) for which the feature space is larger than the batch size. The architectures used for the image datasets are based on the DCGAN (Radford et al., 2016), while results using ResNets are given in Section 4. All images are scaled in between -1 and 1 before running the algorithms. Additional information on the architectures and datasets is given in Appendix. The hyperparameters of other methods are typically chosen as suggested in the authors' reference implementation. The number of unrolling steps in Unrolled GAN is chosen to be 5. For MDGAN, both versions are implemented. The first version, which corresponds to the mode regularizer, has hyperparameters $\lambda_1 = 0.2$ and $\lambda_2 = 0.4$, for the second version, which corresponds to manifold diffusion training for regularized GANs, has $\lambda = 10^{-2}$. WGAN-GP uses $\lambda = 10.0$ and $n_{\mathrm{critic}} = 5$. All models are trained using Adam (Kingma & Ba, 2015) with $\beta_1 = 0.5$, $\beta_2 = 0.999$ and learning rate $10^{-3}$ for both the generator and discriminator. Unless stated otherwise, the batch

size is taken to 256. Examples of random generations of all the GANs are given in Appendix. Notice that in this section we report the results achieved only at the end of the training.

## 3.1 ARTIFICIAL DATA

**Synthetic.** The ring is a mixture of eight two-dimensional spherical Gaussians in the plane with means $2.5 \times (\cos((2\pi/8)i), \sin((2\pi/8)i))$ and std $0.01$ for $i \in \{1, \ldots, 8\}$. The 2D-grid is a mixture of 25 two-dimensional isotropic normals in the plane with means separated by 2 and with standard deviation $0.05$. All models have the same architecture, with $\ell = 256$ following Elfeki et al. (2019), and are trained for 25k iterations. The evaluation is done by sampling 3k points from the generator network. A sample is counted as high quality if it is within 3 standard deviations of the nearest mode. The experiments are repeated 10 times for all models and their performance is compared in Table 1.

BuresGANs consistently capture all the modes and produces the highest quality samples. The training progress of the Alt-BuresGAN is shown on Figure 1, where we observe that all the modes early on in the training procedure, afterwards improving the quality. The training progress of the other GAN models listed in Table 1 is given in Appendix. Although BuresGAN training times are larger than most other methods for this low dimensional example, we show in Appendix D.1 that BuresGAN scales better with the input data dimension and architecture complexity.

| Step 0 | Step 2k | Step 4k | Step 6k | Step 8k | Step 25k |
|--------|---------|---------|---------|---------|----------|

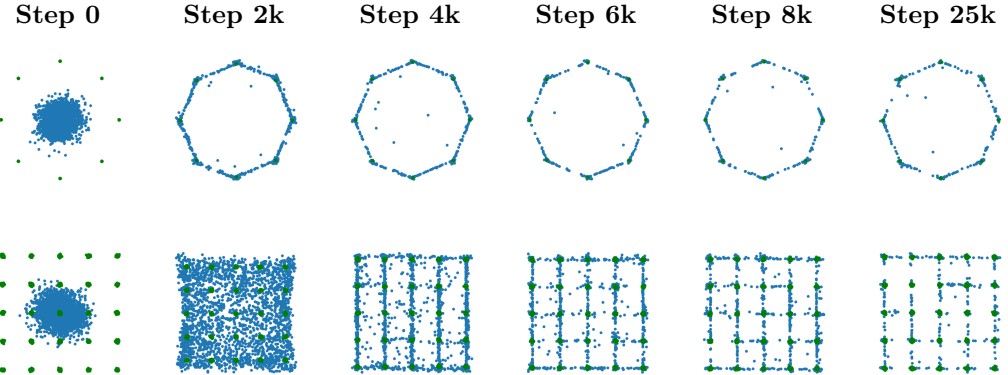

Figure 1: Figure accompanying Table 1, the progress of Alt-BuresGAN on the synthetic examples. Each column shows 3k samples from the training of the generator in blue and 3k samples from the true distribution in green.

|  | Grid with 25 modes | | | Ring with 8 modes | | |
|---|---|---|---|---|---|---|
|  | Nb modes | % in $3\sigma$ | time (s) | Nb modes | % in $3\sigma$ | time (s) |
| GAN | 22.9(4) | 76(13) | 32(0.2) | 7.4(2) | 76(25) | 32(0.4) |
| WGAN-GP | 24.9(0.3) | 77(10) | 225(1) | 7.1(1) | 9(5) | 224(1) |
| MDGAN-v1 | 21(3) | 49(18) | 65(0.2) | 6.5(2) | 65(26) | 65(0.5) |
| MDGAN-v2 | **25**(0) | 68(11) | 81(0.2) | 5(3) | 20(15) | 81(0.8) |
| UnrolledGAN | 19.7(1) | 78(19) | 98(0.2) | **8**(0) | 77(18) | 98(0.7) |
| VEEGAN | **25**(0) | 67(3) | 49(0.3) | **8**(0) | 29(5) | 49(0.5) |
| GDPP | 20.5(5) | 79(23) | 392(25) | 7.5(0.8) | 73(25) | 404(5) |
| PacGAN2 | 23.6(4) | 65(28) | 43(0.4) | **8**(0) | 81(15) | 43(0.6) |
| Alt-BuresGAN | **25**(0) | **84**(1) | 298(0.6) | **8**(0) | **84**(6) | 298(1) |
| BuresGAN | **25**(0) | 82(1) | 279(0.4) | **8**(0) | 82(4) | 278(1) |

Table 1: Experiments on the synthetic datasets with a GPU Nvidia Quadro P4000. Average(std) over 10 runs. All the models are trained for 25k iterations and the total training time is indicated in seconds.

**Stacked MNIST.** The Stacked MNIST dataset is specifically constructed to contain 1000 known modes. This is done by stacking three digits, sampled uniformly at random from the original MNIST dataset, each in a different channel. The BuresGAN models are compared to the other methods and are trained for 25k iterations. The results with different batch sizes can be found in Table 2. For the evaluation of the performance, we follow Metz et al. (2017) and use the following metrics: the number of captured modes measures mode collapse and the KL divergence, which also measures sample quality. The mode of each generated image is identified by using a standard MNIST classifier which is trained up to 98.43% accuracy on the validation set (see Appendix), and classifies each channel of the fake sample. The same classifier is used to count the number of captured modes. The metrics are calculated based on 10k generated images for all the models. Interestingly, for most models, an improvement is observed in the quality of the images – KL divergence – and in terms of mode collapse – number of modes attained – as the size of the batch increases. For the same batch size, architecture and iterations, the image quality is improved by BuresGAN, which is robust with respect to batch size and architecture choice. The other methods show a higher variability over the different experiments. WGAN-GP has the best single run performance with a discriminator with 3 convolutional layers and has on average a superior performance when using a discriminator with 2 convolutional layers (see Table 21 in Appendix) but sometimes fails to converge when increasing the number of discriminator layers by 1 along with increasing the batch size. MDGANv2, VEEGAN, GDPP and WGAN-GP often have an excellent single run performance. However, when increasing the number of discriminator layers, the training of these models has a tendency to collapse more often as indicated by the large std. Vanilla GAN is one of the best performing models in the variant with 3 layers. This indicates that, for certain datasets, careful architecture tuning can be more important than complicated training schemes. A lesson from Table 2 is that BuresGAN's mode coverage does not vary much if the batch size increases, although the KL divergence seems to be slightly improved. Generated samples from the Alt-BuresGAN are given in Figure 2. Since VEEGAN and PacGAN papers use a different setup, we also report in Table 12 the specifications of the different settings.

| Stacked MNIST | CIFAR-10 | CIFAR-100 |
| :---: | :---: | :---: |

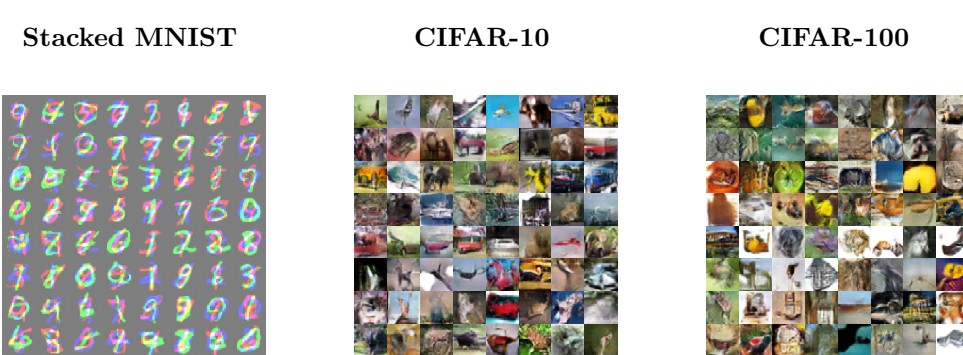

Figure 2: Generated samples from a trained Alt-BuresGAN, with a DCGAN architecture.

As it was observed by previous papers such as in Lin et al. (2018) and Xiao et al. (2018), even the vanilla GAN can achieve an excellent mode coverage on certain architectures. Since the results of Table 2 do not determine a clear winner, we repeated a similar experiment with a more challenging architecture (reported in Table 10) which includes 4 convolutional layers for both the generator and discriminator in an analogous way to the experiment proposed in Srivastava et al. (2017). The leading methods of Table 2 are considered for this experiment. Then, we observe in Table 3 that vanilla GAN and GDPP collapse for this architecture. Compared with the results of Table 2, the difference between the methods is more significant: WGAN-GP yields the best result and is followed by the BuresGAN models. However note that our empirical results indicate that WGAN-GP is sensitive to the choice of architecture and hyperparameters and its training time is also longer as it can be seen from Table 19 in Appendix.

| | | Nb modes(↑) | | | KL div.(↓) | | |
|---|---|---|---|---|---|---|---|
| | Batch size | 64 | 128 | 256 | 64 | 128 | 256 |
| | GAN | 993.3(3.1) | 995.4(1.7) | **998.3**(1.2) | 0.28(0.02) | **0.24**(0.02) | 0.21(0.02) |
| | WGAN-GP | 980.2(57) | 838.3(219) | 785.1(389) | **0.26**(0.34) | 1.05(1) | 1.6(2.4) |
| | MDGAN-v1 | 233.8(250) | 204.0(202) | 215.5(213) | 5.0(1.6) | 4.9(1.3) | 5.0(1.2) |
| 3 conv. layers | MDGAN-v2 | 299.9(457) | 300.4(457) | 200.0(398) | 4.8(3.0) | 4.7(3.0) | 5.5(2.6) |
| | UnrolledGAN | 934.7(107) | 874.1(290) | 884.9(290) | 0.72(0.51) | 0.98(1.46) | 0.90(1.4) |
| | VEEGAN | 974.2(10.3) | 687.9(447) | 395.6(466) | 0.33(0.05) | 2.04(2.61) | 3.52(2.64) |
| | GDPP | 894.2(298) | 897.1(299) | 997.5(1.4) | 0.92(1.92) | 0.88(1.93) | **0.20**(0.02) |
| | PacGAN2 | 989.8(4.0) | 993.3(4.8) | 897.7(299) | 0.33(0.02) | 0.29(0.04) | 0.87(1.94) |
| | BuresGAN | 993.5(2.7) | **996.3**(1.6) | 997.1(2.4) | 0.29(0.02) | 0.25(0.02) | 0.23(0.01) |
| | Alt-BuresGAN | **995.0**(2.4) | 995.6(3.6) | 995.5(2.4) | 0.28(0.03) | 0.27(0.02) | 0.24(0.02) |

Table 2: KL-divergence between the generated distribution and true distribution for an architecture with 3 conv. layers for the Stacked MNIST dataset. The number of counted modes assesses mode collapse. The results are obtained after 25k iterations and we report the average(std) over 10 runs.

| | | Nb modes(↑) | KL div.(↓) |
|---|---|---|---|
| | GAN | 21.6(25.8) | 5.10(0.83) |
| | WGAN-GP | **999.7**(0.6) | **0.11**(0.006) |
| | MDGAN-v2 | 729.5(297.9) | 1.76(1.65) |
| | UnrolledGAN | 24.3(23.61) | 4.96(0.68) |
| 4 conv. layers | VEEGAN | 816.1(269.6) | 1.33(1.46) |
| | GDPP | 33.3(39.4) | 4.92(0.80) |
| | PacGAN2 | 972.4(12.0) | 0.45(0.06) |
| | BuresGAN | 989.9(4.7) | 0.38(0.06) |
| | Alt-BuresGAN | 990.0(4.9) | 0.33(0.04) |

Table 3: Stacked MNIST experiment for an architecture with 4 conv. layers. All the models are trained for 25k iterations with a batch size of 64, a learning rate of $2e{-}4$ for Adam and a normal latent distribution. The evaluation is over 10k samples and we report an average(std) over 10 runs.

## 3.2 REAL IMAGES

**Metrics.** The image quality is assessed thanks to the Inception Score (IS), Fréchet Inception Distance (FID), Inference via Optimization (IvO) and Sliced Wasserstein Distance (SWD). The latter was also used in Elfeki et al. (2019) and Karras et al. (2017) to evaluate the quality of images as well as the severity of mode-collapse. In a word, SWD evaluates the multiscale statistical similarity between distributions of local image patches drawn from Laplacian pyramids. A small Wasserstein distance indicates that the distribution of the patches is similar, thus real and fake images appear similar in both appearance and variation at this spatial resolution. IvO (Metz et al., 2017) measures mode collapse by comparing real images with the nearest generated image. It relies on an optimization procedure within the latent space to find the closest generated image to a given test image, and returns the distance which can be large in the case of mode collapse. The metrics are calculated based on 10k generated images for all the models.

**CIFAR datasets.** In Table 4, we evaluate the performance on the $32 \times 32 \times 3$ CIFAR datasets. While all models are trained for 100k iterations, the best performance is observed for BuresGAN and Alt-BuresGAN in terms of image quality, measured by FID and Inception Score, and in terms of mode collapse, measured by SWD and IvO. We also notice that UnrolledGAN, VEEGAN and WGAN-GP have difficulty converging to a satisfactory result for this architecture. This in contrast to the 'simpler' synthetic data and the Stacked MNIST dataset, where the models get comparable performance to BuresGAN and Alt-BuresGAN. In Arjovsky et al. (2017), WGAN-GP achieves a very good performance on CIFAR-10 with a ResNet architecture which is considerably more complicated than the DCGAN used here. Therefore, results with a Resnet architure are reported in Section 4. Also, for this architecture and number of training iterations, MDGAN-v1 and MDGAN-v2 did not converge to a meaningful result in our simulations.

|  | CIFAR-10 | | | | CIFAR-100 | | | |
| --- | --- | --- | --- | --- | --- | --- | --- | --- |
|  | IvO($\downarrow$) | IS($\uparrow$) | FID($\downarrow$) | SWD($\downarrow$) | IvO($\downarrow$) | IS($\uparrow$) | FID($\downarrow$) | SWD($\downarrow$) |
| GAN | 0.30(0.06) | 5.67(0.22) | 59(8.5) | 3.7(0.9) | 0.37(0.10) | 5.2(1.1) | 91.7(66) | 7.8(4.9) |
| WGAN-GP | 0.49(0.24) | 2.01(0.47) | 291(87) | 8.3(1.9) | 0.54(0.27) | 1.2(0.5) | 283(113) | 9.7(2.5) |
| UnrolledGAN | 0.36(0.08) | 3.1(0.6) | 148(42) | 9.0(5) | 0.44(0.10) | 3.2(0.7) | 172.9(40) | 13.1(9.2) |
| VEEGAN | 0.36(0.08) | 2.5(0.6) | 198(33.5) | 12.0(3) | 0.35(0.08) | 2.8(0.7) | 177.2(27) | 12.8(3.9) |
| GDPP | **0.29**(0.05) | 5.76(0.27) | 62.1(5.5) | 4.1(1.1) | 0.37(0.05) | 5.9(0.2) | 65.0(8) | 4.4(1.9) |
| PacGAN2 | 1.00(0.35) | 5.51(0.18) | 60.61(5.9) | 2.95(1) | 0.97(0.35) | 5.6(0.1) | 59.9(5.2) | 4.0(1.8) |
| BuresGAN | 0.30(0.03) | **6.34**(0.17) | **43.7**(0.9) | 2.1(0.6) | 0.36(0.08) | **6.5**(0.1) | **47.2**(1.2) | 2.1(1.0) |
| Alt-BuresGAN | **0.29**(0.03) | 6.23(0.07) | 45.4(2.8) | **1.7**(0.9) | **0.33**(0.05) | 6.4(0.1) | 49.4(3.4) | **1.8**(0.6) |

Table 4: Generation quality on CIFAR-10 and CIFAR-100 with DCGAN architecture. Average(std) over 10 runs. 100k iterations for each. For improving readability, SWD score was multiplied by 100.

CIFAR-100 dataset consists of 100 different classes and is therefore more diverse. Compared to the original CIFAR-10 dataset, the performance of the proposed GANs remains almost the same, with a small increase in IvO. An exception is vanilla GAN, which shows a higher presence of mode collapse as measured by IvO and SWD.

**STL-10.** The STL-10 dataset includes higher resolution images of size $96 \times 96 \times 3$. The best performing models from previous experiments are trained for 150k iterations. Samples of generated images from a trained Alt-BuresGAN are given on Figure 3. The metrics are calculated based on 5k generated images for all the models. Compared to the previous datasets, GDPP and vanilla GAN are rarely able to generate high quality images on the higher resolution STL-10 dataset. Only BuresGANs are capable of consistently generating high quality images as well as preventing mode collapse, for the same architecture.

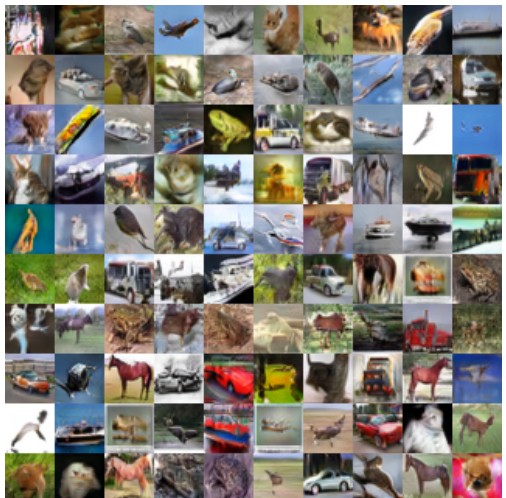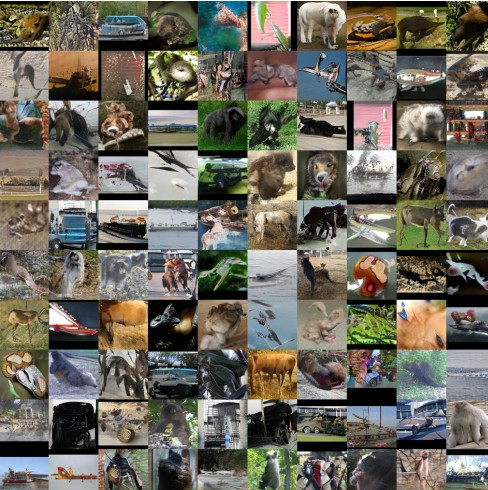

Figure 3: Images generated by BuresGAN with a ResNet architecture for CIFAR-10 (left) and STL-10 (right) datasets. The STL-10 samples are full-sized $96 \times 96$ images.

**Timings.** The computing times for these datasets are in Appendix. For the same number of iterations, (alt-)BuresGAN training time is comparable to WGAN-GP training for the simple data in Table 1. For more complicated architectures, (alt-)BuresGAN scales better and the training time was observed to be significantly shorter with respect to WGAN-GP and several other methods.

# 4 HIGH QUALITY GENERATION USING A RESNET ARCHITECTURE

As noted by Lucic et al. (2018), a fair comparison should involve GANs with the same architecture, and this is why we restricted in our paper to a classical DCGAN architecture. It is natural to

|  | IvO($\downarrow$) | IS($\uparrow$) | FID($\downarrow$) | SWD($\downarrow$) |
|---|---|---|---|---|
| GAN | 0.50(0.15) | 2.9(1.8) | 237(54) | 12.3(4.1) |
| GDPP | 0.46(0.09) | 3.3(2.2) | 232(84) | 8.2(4.0) |
| PacGAN2 | 0.97(0.05) | 4.7(1.5) | 161(36) | 8.1(4.3) |
| BuresGAN | **0.44**(0.05) | **7.6**(0.3) | **109**(7) | **2.3**(0.3) |
| Alt-BuresGAN | 0.45(0.04) | 7.5(0.3) | 110(4) | 2.8(0.4) |

Table 5: Generation quality on STL-10 with DCGAN architecture. Average(std) over 5 runs. 150k iterations for each. SWD score was multiplied by 100 for improving readability.

question the performance of BuresGAN with a ResNet architecture. Hence, we trained BuresGAN on the CIFAR-10 and STL-10 datasets by using the ResNet architecture taken from Gulrajani et al. (2017). In this section, the STL-10 images are rescaled to $48 \times 48 \times 3$ according the procedure described in Miyato et al. (2018); Lee et al. (2019); Wang et al. (2019), so that the comparison of IS and FID scores with other works is meaningful. Note that BuresGAN has no parameters to tune, except for the hyperparameters of the optimizers. The results are displayed in Table 6, where the scores of state-of-the-art unconditional GAN models with a ResNet architecture are also reported. In contrast with Section 3, we report here the best performance achieved at any time during the training, averaged over several runs. To the best of our knowledge, our method achieves a new state of the art inception score on STL-10 and is within a standard deviation of state of the art on CIFAR-10 using a ResNet architecture. The FID score achieved by BuresGAN is nonetheless smaller than the reported FID scores for GANs using a ResNet architecture. A visual inspection of the generated images in Figure 3 shows that the high inception score is warranted, the samples are clear, diverse and often recognizable. BuresGAN also performs well on the full-sized STL-10 data set where an inception score of $11.11 \pm 0.19$ and an FID of $50.9 \pm 0.13$ is achieved (average and std over 3 runs).

|  | CIFAR-10 | | STL-10 | |
|---|---|---|---|---|
|  | IS ($\uparrow$) | FID ($\downarrow$) | IS ($\uparrow$) | FID ($\downarrow$) |
| WGAN-GP ResNet (Gulrajani et al., 2017)* | 7.86(0.07) | 18.8[†] | / | / |
| InfoMax-GAN (Lee et al., 2019)* | 8.08(0.08) | 17.14(0.20) | 8.54(0.12) | 37.49(0.05) |
| SN-GAN ResNet (Miyato et al., 2018)* | 8.22(0.05) | 21.7(0.21) | 9.10(0.04) | 40.1(0.50) |
| ProgressiveGAN (Karras et al., 2017)* | 8.80(0.05) | / | / | / |
| CR-GAN (Zhang et al., 2020)* | 8.4 | 14.56 | / | / |
| NCSN (Song & Ermon, 2019)* | **8.87**(0.12) | 25.32 | / | / |
| Improving MMD GAN (Wang et al., 2019)* | 8.29 | 16.21 | 9.34 | 37.63 |
| WGAN-div (Wu et al., 2018)* | / | 18.1[†] | / | / |
| BuresGAN ResNet (Ours) | 8.81(0.08) | **12.91**(0.40) | **9.67**(0.19) | **31.42**(1.01) |

Table 6: Best achieved IS and FID, using a ResNet architecture. Results with an asterisk are quoted from their respective papers (std in parenthesis). BuresGAN results were obtained after $300k$ iterations and averaged over 3 runs. The result indicated with † are taken from Wu et al. (2018). For all the methods, the STL-10 images are rescaled to $48 \times 48 \times 3$ in contrast with Table 5.

## 5    CONCLUSION

In this work, we discussed an additional term based on the Bures distance whichpromotes a matching of the distribution of the generated and real data in a feature space $\mathbb{R}^f$. The Bures distance admits both a feature space and kernel based expression, which makes the proposed model time and data efficient when compared to state of the art models. Two training procedures are proposed: Algorithm 1 deals with the squared Bures distance as an additive term to the generator loss, while an alternating training is used in Algorithm 2 so that no extra parameter is introduced. Our experiments show that the proposed methods are capable of reducing mode collapse and, on the real datasets, achieve a clear improvement of sample quality without parameter tuning and without the need for regularization such as a gradient penalty. Moreover, the proposed GANs show a stable performance over different architectures, datasets and hyperparmeters.

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

## A  PROOFS

*Proof of Lemma 1.* (i) is a consequence of Corollary 2.3 in (Hong & Horn, 1991). (ii) We now follow (Oh et al., 2020). Thanks to (i), we have $AB = PDP^{-1}$ where $D$ is a nonnegative diagonal and the columns of $P$ contain the right eigenvectors of $AB$. Therefore, $\text{Tr}((AB)^{1/2}) = \text{Tr}(D^{1/2})$. Then, $YAY^\top$ is clearly diagonalizable. Let us show that it shares its nonzero eigenvalues with $AB$. a) We have $ABP = PD$, so that, by multiplying of the left by $Y$, it holds that $(YAY^\top)YP = YPD$. b) Similarly, suppose that we have the eigenvalue decomposition $YAY^\top Q = Q\Lambda$. Then, we have $BAY^\top Q = Y^\top Q\Lambda$ with $B = Y^\top Y$. This means that the non-zero eigenvalues of $YAY^\top$ are also eigenvalues of $BA$. Since $A$ and $B$ are symmetric, this completes the proof.  □

*Proof of Lemma 2.* The result follows from Lemma 1 and its proof, where $A = X^\top X$ and $B = Y^\top Y$.  □

## B  DETAILS OF THE THEORETICAL RESULTS

Let $A$ and $B$ be symmetric and positive semi-definite matrices. Let $A^{1/2} = U \operatorname{diag}(\sqrt{\lambda})U^\top$ where $U$ and $\lambda$ are obtained thanks to the eigenvalue decomposition $A = U \operatorname{diag}(\lambda)U^\top$. We show here that the Bures distance between $A$ and $B$ is

$$\mathcal{B}(A,B)^2 = \min_{U \in O(\ell)} \|A^{1/2} - B^{1/2}U\|_F^2 = \text{Tr}(A + B - 2(A^{\frac{1}{2}}BA^{\frac{1}{2}})^{\frac{1}{2}}), \tag{6}$$

where $O(\ell)$ is the set of $\ell \times \ell$ orthonormal matrices. We can simplify the above expression as follows

$$\min_{U \in O(\ell)} \|A^{1/2} - B^{1/2}U\|_F^2 = \text{Tr}(A) + \text{Tr}(B) - 2 \max_{U \in O(\ell)} \text{Tr}(A^{1/2}B^{1/2}U) \tag{7}$$

since $\text{Tr}(U^\top B^{1/2}A^{1/2}) = \text{Tr}(A^{1/2}B^{1/2}U)$. Let the characteristic function of the set of orthonormal matrices be $f(U) = \chi_{O(\ell)}(U)$ that is, $f(U) = 0$ if $U \in O(\ell)$ and $+\infty$ otherwise.

**Lemma 3.** *The Fenchel conjugate of $f(U) = \chi_{O(\ell)}(U)$ is $f^\star(M) = \|M\|_\star$, where the nuclear norm is $\|M\|_\star = \text{Tr}(\sqrt{M^\top M})$ and $U, M \in \mathbb{R}^{\ell \times \ell}$.*

*Proof.* The definition of the Fenchel conjugate with respect to the Frobenius inner product gives

$$f^\star(M) = \sup_{U \in \mathbb{R}^{\ell \times \ell}} \text{Tr}(U^\top M) - f(U) = \max_{U \in O(\ell)} \text{Tr}(U^\top M).$$

Next we decompose $M$ as follows: $M = W\Sigma V^\top$, where $W, V \in O(\ell)$ are orthogonal matrices and $\Sigma$ is a $\ell \times \ell$ diagonal matrix with non negative diagonal entries, such that $MM^\top = W\Sigma^2 W^\top$ and $M^\top M = V\Sigma^2 V^\top$. Notice that the non zero diagonal entries of $\Sigma$ are the singular values of $M$. Then,

$$\max_{U \in O(\ell)} \text{Tr}(U^\top M) = \max_{U \in O(\ell)} \text{Tr}(W\Sigma V^\top U) = \max_{U' \in O(\ell)} \text{Tr}(\Sigma U'),$$

where we renamed $U' = V^\top UW$. Next, we remark that $\text{Tr}(\Sigma U') = \text{Tr}(\Sigma \operatorname{diag}(U'))$. Since by construction, $\Sigma$ is diagonal with non negative entries the maximum is attained at $U' = \mathbb{I}$. Then, the optimal objective is $\text{Tr}(\Sigma) = \text{Tr}(\sqrt{M^\top M})$.  □

By taking $M = A^{1/2}B^{1/2}$ we obtain equation 6. Notice that the role of $A$ and $B$ can be exchanged in equation 6 since $U$ is orthogonal.

# C TRAINING DETAILS

## C.1 SYNTHETIC ARCHITECTURES

Following the recommendation in the original work (Srivastava et al., 2017), the same fully-connected architecture is used for the VEEGAN reconstructor in all experiments.

| Layer | Output | Activation |
|-------|--------|------------|
| Input | 256 | - |
| Dense | 128 | tanh |
| Dense | 128 | tanh |
| Dense | 2 | - |

| Layer | Output | Activation |
|-------|--------|------------|
| Input | 2 | - |
| Dense | 128 | tanh |
| Dense | 128 | tanh |
| Dense | 1 | - |

Table 7: The generator and discriminator architectures for the synthetic examples.

| Layer | Output | Activation |
|-------|--------|------------|
| Input | 2 | - |
| Dense | 128 | tanh |
| Dense | 256 | - |

| Layer | Output | Activation |
|-------|--------|------------|
| Input | 2 | - |
| Dense | 128 | tanh |
| Dense | 128 | tanh |
| Dense | 256 | tanh |
| Normal | 256 | - |

Table 8: Respectively the MDGAN encoder model and VEEGAN stochastic inverse generator architectures for the synthetic examples. The output of the VEEGAN models are samples drawn from a normal distribution with scale 1 and where the location is learned.

## C.2 STACKED MNIST ARCHITECTURES

| Layer | Output | Activation | BN |
|-------|--------|------------|-----|
| Input | 100 | - | - |
| Dense | 12544 | ReLU | Yes |
| Reshape | 7, 7, 256 | - | - |
| Conv' | 7, 7, 128 | ReLU | Yes |
| Conv' | 14, 14, 64 | ReLU | Yes |
| Conv' | 28, 28, 3 | ReLU | Yes |

| Layer | Output | Activation | BN |
|-------|--------|------------|-----|
| Input | 28, 28, 3 | - | - |
| Conv | 14, 14, 64 | Leaky ReLU | No |
| Conv | 7, 7, 128 | Leaky ReLU | Yes |
| Conv | 4, 4, 256 | Leaky ReLU | Yes |
| Flatten | - | - | - |
| Dense | 1 | - | - |

Table 9: The generator and discriminator architectures for the Stacked MNIST experiments. The BN column indicates whether batch normalization is used after the layer or not. For the experiments with 2 convolution layers in Table 21, the final convolution layer is removed in the discriminator.

| Layer | Output | Activation | BN | | Layer | Output | Activation | BN |
|---|---|---|---|---|---|---|---|---|
| Input | 100 | - | - | | Input | 28, 28, 3 | - | - |
| Dense | 2048 | ReLU | Yes | | Conv | 14, 14, 64 | Leaky ReLU | No |
| Reshape | 2, 2, 512 | - | - | | Conv | 7, 7, 128 | Leaky ReLU | Yes |
| Conv' | 4, 4, 256 | ReLU | Yes | | Conv | 4, 4, 256 | Leaky ReLU | Yes |
| Conv' | 7, 7, 128 | ReLU | Yes | | Conv | 2, 2, 512 | Leaky ReLU | Yes |
| Conv' | 14, 14, 64 | ReLU | Yes | | Flatten | - | - | - |
| Conv' | 28, 28, 3 | ReLU | Yes | | Dense | 1 | - | - |

Table 10: The generator and discriminator architectures for the Stacked MNIST experiments with 4 layers for both generator and discrimintor. The BN column indicates whether batch normalization is used after the layer or not.

| Layer | Output | Activation | BN |
|---|---|---|---|
| Input | 28, 28, 3 | - | - |
| Conv | 14, 14, 3 | ReLU | Yes |
| Conv | 7, 7, 64 | ReLU | Yes |
| Conv | 7, 7, 128 | ReLU | Yes |
| Flatten | - | - | - |
| Dense | 100 | - | - |

Table 11: The MDGAN encoder model architecture for the Stacked MNIST experiments. The BN column indicates whether batch normalization is used after the layer or not.

| Parameters | BuresGAN | PacGAN | VEEGAN | GDPP |
|---|---|---|---|---|
| learning rates | $1 \times 10^{-3}$ & $2 \times 10^{-4}$ | $2 \times 10^{-4}$ | $2 \times 10^{-4}$ | $1 \times 10^{-4}$ ** |
| learning rate decay | no | no | no | no** |
| Adam $\beta_1$ | 0.5 | 0.5 | 0.5 | 0.5 |
| Adam $\beta_2$ | 0.999 | 0.999 | 0.999 | 0.9 |
| iterations | 25000 | 20000 | ? | 15000* |
| disc. conv. layers | 3 & 4 | 4 | 4 | 3 |
| gen. conv. layers | 3 & 4 | 4 | 4 | 3 |
| batch size | 64, 128, 256 & 64 | 64 | 64 | 64 |
| evaluation samples | 10000 | 26000 | 26000 | 26000 |
| $\ell$ (i.e., z dimension) | 100 | 100 | 100 | 128 |
| z distribution | $\mathcal{N}(0, \mathbb{I}_\ell)$ | $\text{unif}[-1, 1]^\ell$ | $\text{unif}[-1, 1]^\ell$ ** | $\text{unif}[-1, 1]^\ell$ |

Table 12: The hyperparameters and architectures of the stacked MNIST experiment performed by the different methods are given here. This represents our best efforts at providing a comparison between the different parameters used. The asterisks ** indicate that the parameters where obtained by the Github repository. Notice that GDPP paper also used 30000 iterations for training DCGAN and unrolled GAN (indicated by *). BuresGAN's column refers to the settings of experiment of Table 2 and Table 3 respectively, for which the different values are separated by the symbol &.

## C.3 CIFAR-10 AND 100 DCGAN ARCHITECTURES

| Layer | Output | Activation | BN | Layer | Output | Activation | BN |
|---|---|---|---|---|---|---|---|
| Input | 100 | - | - | Input | 32, 32, 3 | - | - |
| Dense | 16384 | ReLU | Yes | Conv | 16, 16, 64 | Leaky ReLU | No |
| Reshape | 8, 8, 256 | - | - | Conv | 8, 8, 128 | Leaky ReLU | Yes |
| Conv' | 8, 8, 128 | ReLU | Yes | Conv | 4, 4, 256 | Leaky ReLU | Yes |
| Conv' | 16, 16, 64 | ReLU | Yes | Flatten | - | - | - |
| Conv' | 32, 32, 3 | ReLU | Yes | Dense | 1 | - | - |

Table 13: The generator and discriminator architectures for the CIFAR-10 and CIFAR-100 experiments. The BN column indicates whether batch normalization is used after the layer or not.

| Layer | Output | Activation | BN |
|---|---|---|---|
| Input | 32, 32, 3 | - | - |
| Conv | 16, 16, 3 | ReLU | Yes |
| Conv | 8, 8, 64 | ReLU | Yes |
| Conv | 8, 8, 128 | ReLU | Yes |
| Flatten | - | - | - |
| Dense | 100 | - | - |

Table 14: The MDGAN encoder model architecture for the CIFAR-10 and CIFAR-100 experiments. The BN column indicates whether batch normalization is used after the layer or not.

## C.4 STL-10 DCGAN ARCHITECTURES

| Layer | Output | Activation | BN | Layer | Output | Activation | BN |
|---|---|---|---|---|---|---|---|
| Input | 100 | - | - | Input | 96, 96, 3 | - | - |
| Dense | 36864 | ReLU | Yes | Conv | 48, 48, 64 | Leaky ReLU | No |
| Reshape | 12, 12, 256 | - | - | Conv | 24, 24, 128 | Leaky ReLU | Yes |
| Conv' | 12, 12, 256 | ReLU | Yes | Conv | 12, 12, 256 | Leaky ReLU | Yes |
| Conv' | 24, 24, 128 | ReLU | Yes | Conv | 6, 6, 512 | Leaky ReLU | Yes |
| Conv' | 48, 48, 64 | ReLU | Yes | Flatten | - | - | - |
| Conv' | 96, 96, 3 | ReLU | Yes | Dense | 1 | - | - |

Table 15: The generator and discriminator architectures for the STL-10 experiments. The BN column indicates whether batch normalization is used after the layer or not.

| Layer | Output | Activation | BN |
|---|---|---|---|
| Input | 96, 96, 3 | - | - |
| Conv | 48, 48, 3 | ReLU | Yes |
| Conv | 24, 24, 64 | ReLU | Yes |
| Conv | 12, 12, 128 | ReLU | Yes |
| Conv | 12, 12, 256 | ReLU | Yes |
| Flatten | - | - | - |
| Dense | 100 | - | - |

Table 16: The MDGAN encoder model architecture for the STL-10 experiments. The BN column indicates whether batch normalization is used after the layer or not.

## C.5 RESNET ARCHITECTURES

For CIFAR-10, we used the ResNet architecture from the appendix of Gulrajani et al. (2017) with minor changes as given in Table 17. We used an initial learning rate of $5e-4$ for CIFAR-10 and STL-10. For both datasets, the models are run for 200k iterations. For STL-10, we used a similar architecture that is given in Table 18.

| Layer | Kernel Size | Resample | Output Shape |
|---|---|---|---|
| Input | - | - | 128 |
| Dense | - | - | $200 \cdot 4 \cdot 4$ |
| Reshape | - | - | $200 \times 4 \times 4$ |
| ResBlock | $[3 \times 3] \times 2$ | up | $200 \times 8 \times 8$ |
| ResBlock | $[3 \times 3] \times 2$ | up | $200 \times 16 \times 16$ |
| ResBlock | $[3 \times 3] \times 2$ | up | $200 \times 32 \times 32$ |
| Conv, tanh | $3 \times 3$ | - | $3 \times 32 \times 32$ |

| Layer | Kernel Size | Resample | Output Shape |
|---|---|---|---|
| ResBlock | $[3 \times 3] \times 2$ | Down | $200 \times 16 \times 16$ |
| ResBlock | $[3 \times 3] \times 2$ | Down | $200 \times 8 \times 8$ |
| ResBlock | $[3 \times 3] \times 2$ | Down | $200 \times 4 \times 4$ |
| ResBlock | $[3 \times 3] \times 2$ | - | $200 \times 4 \times 4$ |
| ResBlock | $[3 \times 3] \times 2$ | - | $200 \times 4 \times 4$ |
| ReLu, meanpool | - | - | 200 |
| Dense | - | - | 1 |

Table 17: The generator (top) and discriminator (bottom) ResNet architectures for the CIFAR-10 experiments.

| Layer | Kernel Size | Resample | Output Shape |
|---|---|---|---|
| Input | - | - | 128 |
| Dense | - | - | $200 \cdot 6 \cdot 6$ |
| Reshape | - | - | $200 \times 6 \times 6$ |
| ResBlock | $[3 \times 3] \times 2$ | up | $200 \times 12 \times 12$ |
| ResBlock | $[3 \times 3] \times 2$ | up | $200 \times 24 \times 24$ |
| ResBlock | $[3 \times 3] \times 2$ | up | $200 \times 48 \times 48$ |
| Conv, tanh | $3 \times 3$ | - | $3 \times 48 \times 48$ |

| Layer | Kernel Size | Resample | Output Shape |
|---|---|---|---|
| ResBlock | $[3 \times 3] \times 2$ | Down | $200 \times 24 \times 24$ |
| ResBlock | $[3 \times 3] \times 2$ | Down | $200 \times 12 \times 12$ |
| ResBlock | $[3 \times 3] \times 2$ | Down | $200 \times 6 \times 6$ |
| ResBlock | $[3 \times 3] \times 2$ | - | $200 \times 6 \times 6$ |
| ResBlock | $[3 \times 3] \times 2$ | - | $200 \times 6 \times 6$ |
| ReLu, meanpool | - | - | 200 |
| Dense | - | - | 1 |

Table 18: The generator (top) and discriminator (bottom) ResNet architectures for the STL-10 experiments. For the experiment with full-sized 96x96 images, an extra upsampling block was added to the generator.

# D  ADDITIONAL EXPERIMENTS

## D.1  TIMINGS

The timings per iteration for the experiments presented in the paper are listed in Table 19. Times are given for all the methods considered, although some method do not always generate meaningful images for all datasets. They are measured for 50 iterations after the first 5 iterations, and the average number of iterations per second is computed. The fastest method is the vanilla GAN. BuresGAN has a similar computation cost as GDPP. We observe that (alt-)BuresGAN is significantly faster compared to WGAN-GP. In order to obtain reliable timings, these results were obtained on the same GPU Nvidia Quadro P4000, although, for convenience, the experiments on these image datasets were executed on a machine equipped with different GPUs.

| | stacked MNIST | CIFAR-10 | CIFAR-100 | STL-10 |
|---|---|---|---|---|
| GAN | **0.54**(0.0005) | **0.65**(0.02) | **0.64**(0.0008) | **6.00**(0.01) |
| WGAN-GP | 2.99(0.004) | 3.41(0.009) | 3.41(0.006) | 36.5(0.03) |
| UnrolledGAN | 1.90(0.002) | 2.17(0.003) | 2.18(0.004) | 21.99(0.06) |
| MDGAN-v1 | 1.24(0.002) | 1.47(0.001) | 1.47(0.002) | 13.35(0.03) |
| MDGAN-v2 | 1.66(0.002) | 1.98(0.002) | 1.98(0.002) | 18(0.03) |
| VEEGAN | 0.56(0.006) | 0.66 (0.006) | 0.65(0.004) | 6.10(0.03) |
| GDPP | 0.69(0.02) | 0.80(0.02) | 0.80(0.02) | 7.46(0.03) |
| PacGAN2 | 0.77(0.006) | 0.91(0.007) | 0.91(0.007) | 8.02(0.008) |
| BuresGAN | 0.72(0.02) | 0.82(0.001) | 0.82(0.0008) | 7.6(0.03) |
| Alt-BuresGAN | 0.98(0.008) | 1.15(0.007) | 1.15(0.007) | 10.10(0.03) |

Table 19: Average time per iteration in seconds for the convolutional architecture. Averaged over 5 runs, with std in parenthesis. The batch size is 256. For Stacked MNIST, we use a discriminator architecture with 3 convolutional layers.

## D.2 BEST INCEPTION SCORES ACHIEVED WITH DCGAN ARCHITECTURE

The inception scores for the best trained models are listed in Table 20. For the CIFAR datasets, the largest inception score is significantly better than the mean for UnrolledGAN and VEEGAN. This is the same for GAN and GDPP on the STL-10 dataset, where the methods often converge to bad results. Only the proposed methods are capable of consistently generating high quality images over all datasets.

|              | CIFAR-10 | CIFAR-100 | STL-10 |
| ------------ | -------- | --------- | ------ |
| GAN          | 5.92     | 6.33      | 6.13   |
| WGAN-GP      | 2.54     | 2.56      | /      |
| UnrolledGAN  | 4.06     | 4.14      | /      |
| VEEGAN       | 3.51     | 3.85      | /      |
| GDPP         | 6.21     | 6.32      | 6.05   |
| BuresGAN     | 6.69     | 6.67      | 7.94   |
| Alt-BuresGAN | 6.40     | 6.48      | 7.88   |

Table 20: Inception Score for the best trained models on CIFAR-10, CIFAR-100 and STL-10, with a DCGAN architecture (higher is better).

## D.3 INFLUENCE OF THE NUMBER OF CONVOLUTIONAL LAYERS FOR DCGAN ARCHITECTURE

Also, we provide in Figure 21 results with a DCGAN architecture including only 2 conv. layers for the discriminator in contrast to Table 2 which uses 3 conv. layers.

|              |              | Nb modes($\uparrow$) | | | KL div.($\downarrow$) | | |
| ------------ | ------------ | ----------- | ----------- | ----------- | ----------- | ----------- | ----------- |
|              | Batch size   | 64          | 128         | 256         | 64          | 128         | 256         |
| 2 conv. layers | GAN          | 970.5(5.8)  | 972.7(6.4)  | 979(3.5)    | 0.47(0.04)  | 0.44(0.02)  | 0.41(0.03)  |
|              | WGAN-GP      | **996.7**(1.6) | **997.5**(0.9) | **998.1**(1.5) | **0.25**(0.01) | **0.22**(0.01) | **0.21**(0.05) |
|              | MDGAN-v1     | 115.9(197)  | 260.9(267)  | 134.3(157)  | 5.5(1.4)    | 4.9(1.7)    | 5.8(0.9)    |
|              | MDGAN-v2     | 698.1(456)  | 898.4(299)  | 599.2(488)  | 2.2(3.0)    | 0.86(1.9)   | 2.8(3.2)    |
|              | UnrolledGAN  | 953.5(11)   | 971.4(4.8)  | 966.2(17.3) | 0.71(0.06)  | 0.60(0.04)  | 0.58(0.10)  |
|              | VEEGAN       | 876.7(290)  | 688.5(443)  | 775.9(386)  | 0.92(1.6)   | 1.9(2.4)    | 1.54(2.2)   |
|              | GDPP         | 974.4(3.3)  | 978.2(7.6)  | 980.5(6.0)  | 0.45(0.02)  | 0.43(0.03)  | 0.41(0.03)  |
|              | PacGAN2      | 969.8(6.9)  | 971.1(3.6)  | 977.9(4.3)  | 0.54(0.04)  | 0.51(0.01)  | 0.48(0.02)  |
|              | BuresGAN     | 973.2(1.3)  | 979.9(4.0)  | 981.1(4.9)  | 0.36(0.02)  | 0.30(0.02)  | 0.25(0.01)  |
|              | Alt-BuresGAN | 975.4(6.8)  | 978.2(5.4)  | 980.2(3.0)  | 0.37(0.02)  | 0.30(0.01)  | 0.28(0.01)  |

Table 21: KL-divergence between the generated distribution and true distribution (Quality, lower is better). The number of counted modes indicates the amount of mode collapse (higher is better). 25k iterations and average and std over 10 runs. Same architecture as in Table 2 with a discriminator with 2 convolutional layers.

# E ADDITIONAL FIGURES

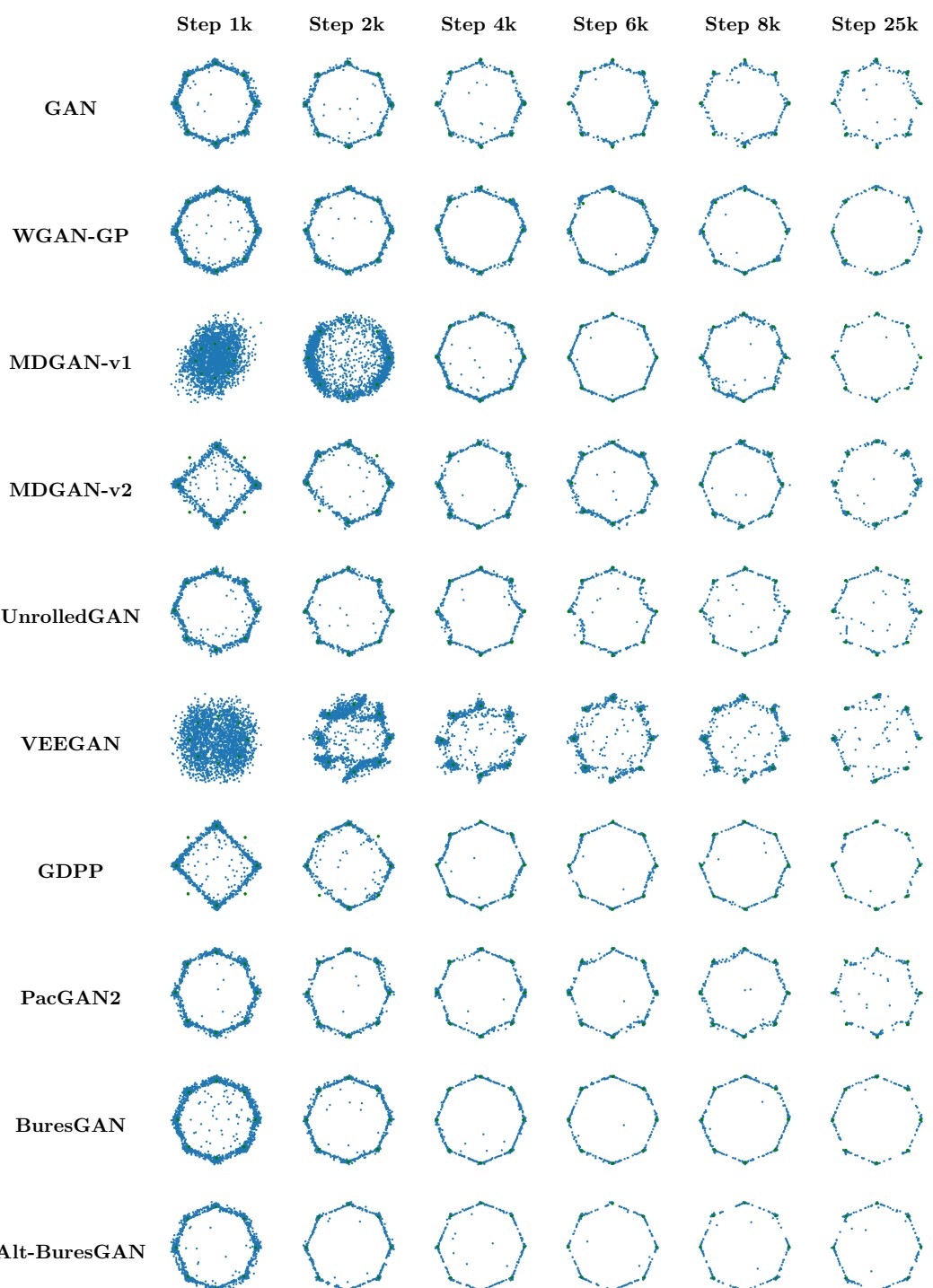

Figure 4: The progress of different GANs on the synthetic ring example. Each column show 3000 samples from the training generator in blue with 3000 samples from the true distribution in green.

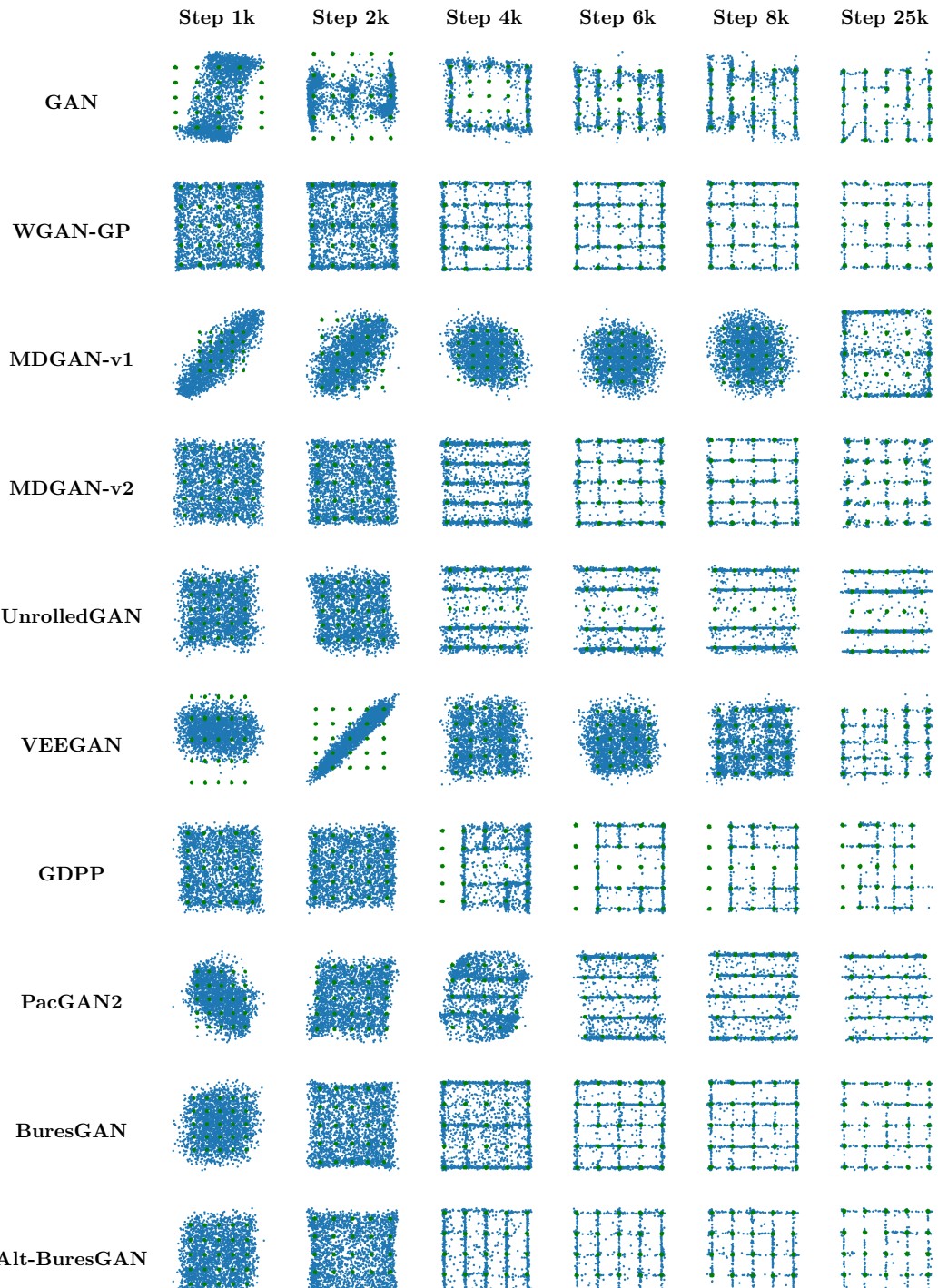

Figure 5: The progress of different GANs on the synthetic grid example. Each column show 3000 samples from the training generator in blue with 3000 samples from the true distribution in green.

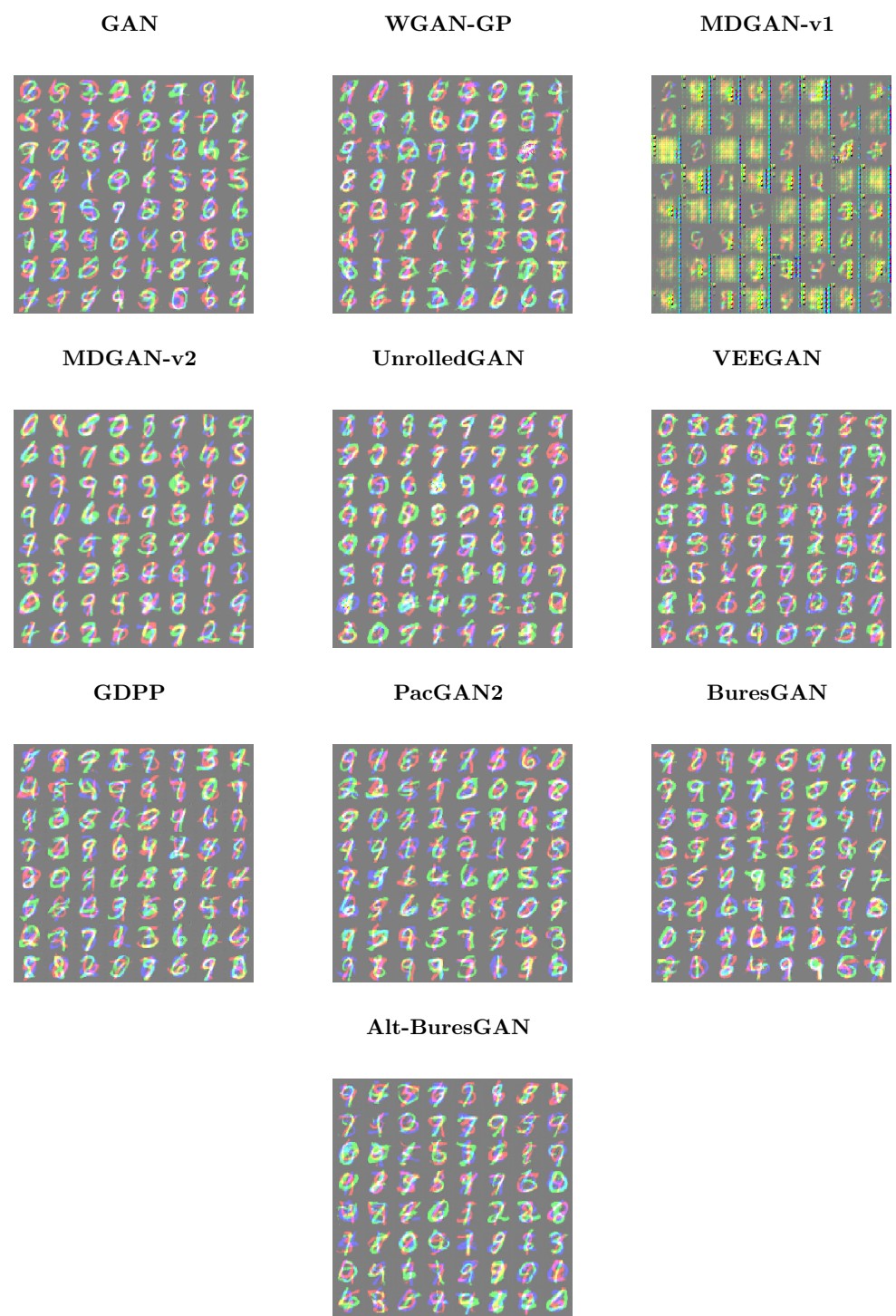

Figure 6: Generated images for the Stacked MNIST dataset. Each model is trained with 3 layers and mini-batch size 256. Each square shows 64 samples from the trained generator.

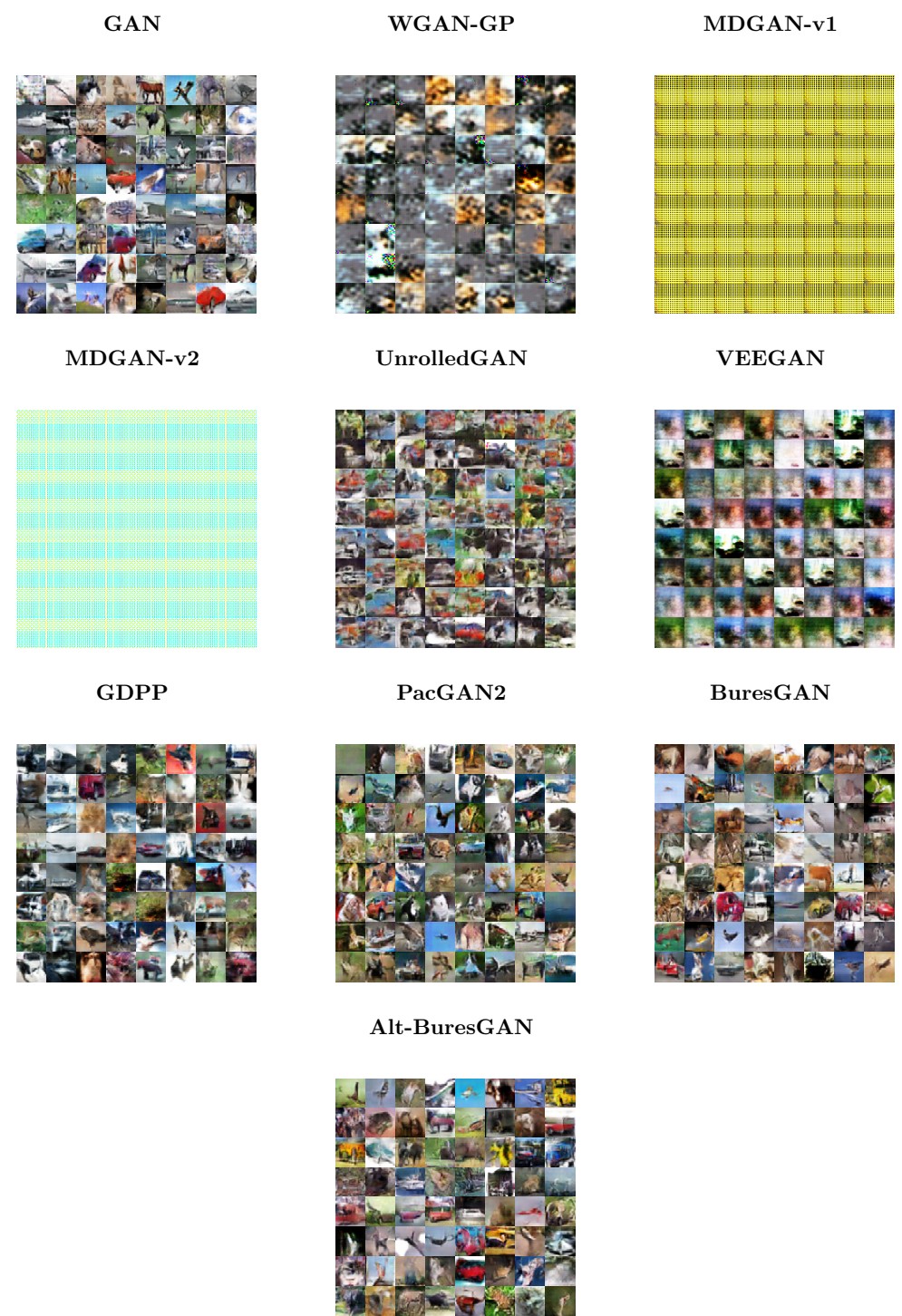

Figure 7: Generated images for CIFAR-10 using a DCGAN architecture. Each square shows 64 samples from the trained generator.

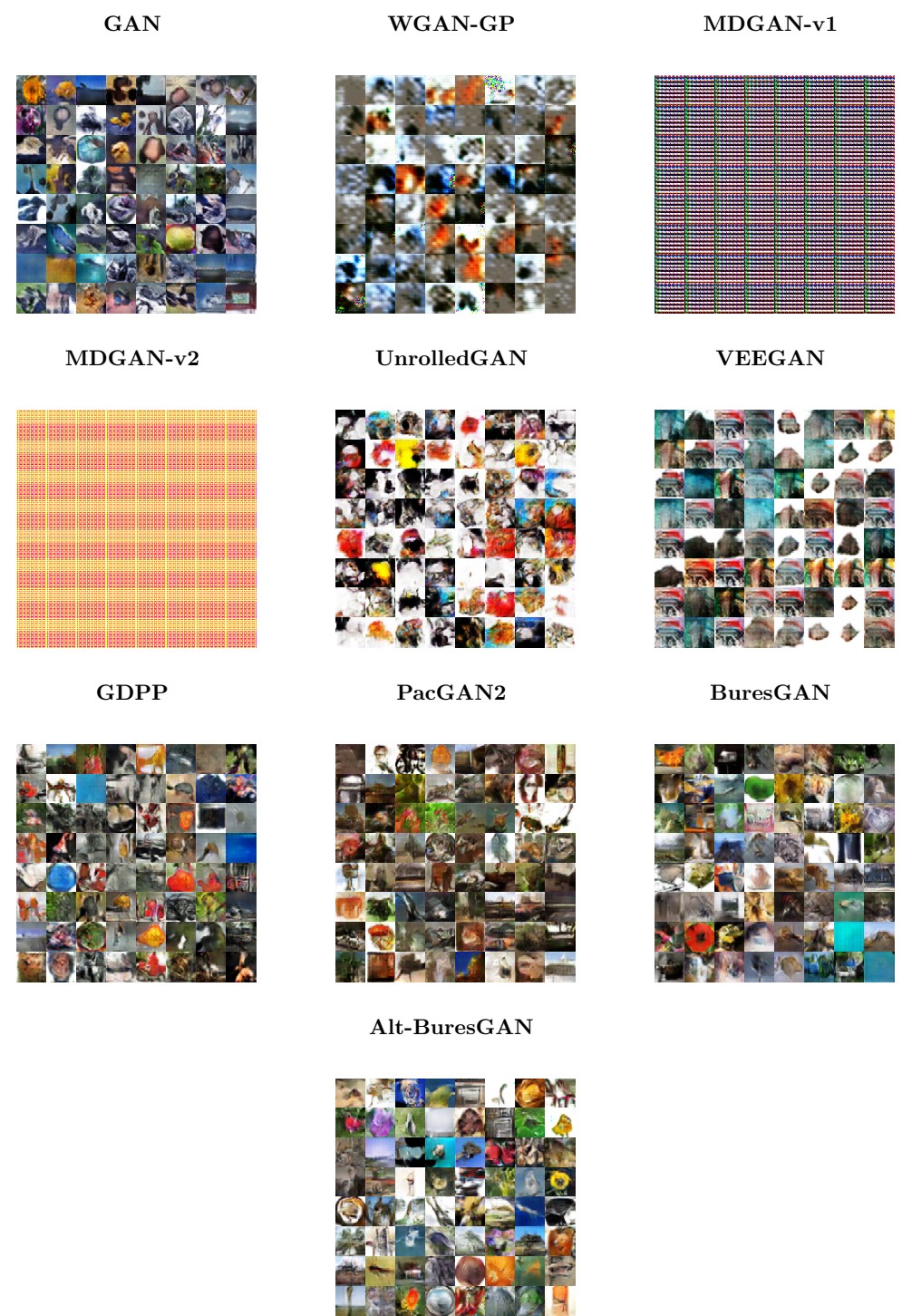

Figure 8: Generated images for CIFAR-100 using a DCGAN architecture. Each square shows 64 samples from the trained generator.

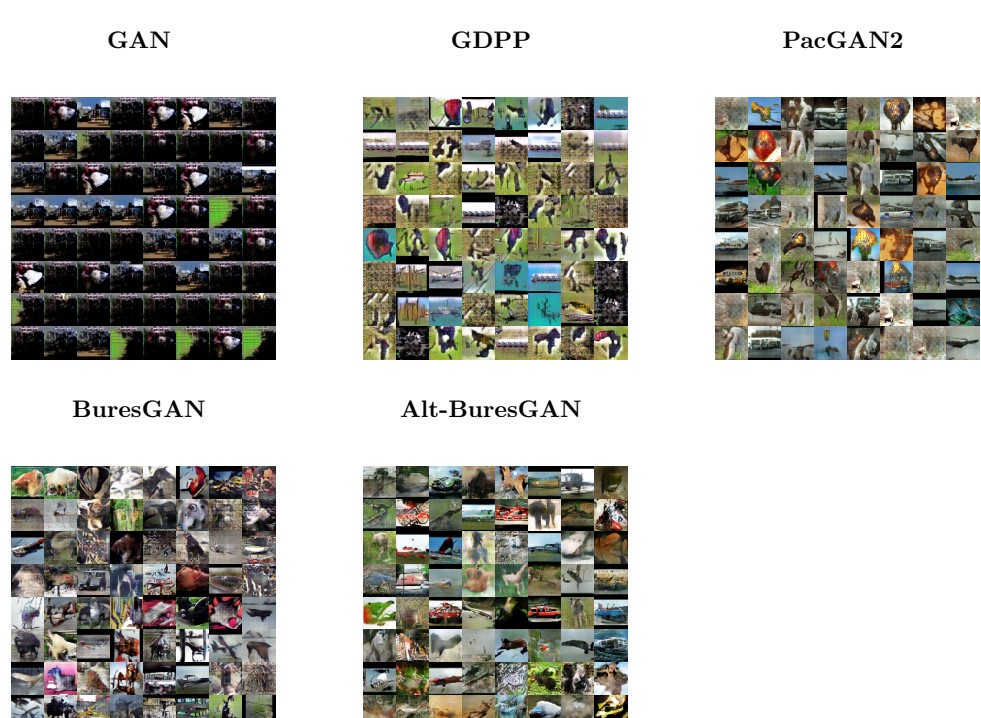

Figure 9: Generated images for STL-10 using a DCGAN architecture. Each square shows 64 samples from the trained generator.

