# OpenReview forum: "The Bures Metric for Taming Mode Collapse in Generative Adversarial Networks"
_ICLR.cc/2021/Conference — Reject_

### Official Review · AnonReviewer2 · 2020-10-24
**Review on THE BURES METRIC FOR TAMING MODE COLLAPSE IN GENERATIVE ADVERSARIAL NETWORKS (revised)**

**Rating:** 3
**Confidence:** 4

**Review:**



Summary: This paper proposes to use the squared Bures distance in discriminator feature space to match the generated and real distributions. This proposed method does not require any modification to the network architectures, and is easy to implement. The proposed method produces good empirical results with simple generator architectures in synthetic and real datasets.

Reason for score: While I find strong interest in the proposed method, the way in which it is presented in this paper does not permit me to fairly judge the merits of this method. The experiments are not thorough and the quality of writing is subpar. Thus, I vote for reject on this paper.

Pros.
This method introduced in this paper is theoretically sound. The idea of imposing additional distribution matching in the feature space of the discriminator is an interesting direction that should warrant more studies. Extending beyond the current scope, the use of the adversarial network as both a discriminator and a feature extractor has great potential as a research direction.

Cons
1. The choice of synthetic experiments may be too easy to discern difference between methods. Most GANs get similar results on the 2D grid experiments as the bottleneck is mostly in the modelling capacity of the generator at expressing non-linearities. Previous works have also achieved perfect mode coverage on stacked-mnist with similar network architecture (https://arxiv.org/pdf/1712.04086.pdf).
2.The Bures metric is one among many metrics for comparing covariance matrices. Ablation on the choice of distance metric should be performed.
3. Higher performing methods with similar network capacities have been left out in the performance tables. This is counter-productive to the purpose of establishing context. Very few recent methods evaluate DCGAN on cifar10 or STL as the architecture is too limiting for these complex datasets. Only inception scores are reported on resnet experiments. Thorough experimentation and evaluation with a modern architecture on RGB datasets would greatly help the case for this paper.

4. The writing in this paper requires significant rework to reach publication quality. Grammatical mistakes and convoluted sentences are too frequent and significantly detracts from the idea being presented.

Minor comments
“...images, although, the training...” run-on sentence
“; and a discriminator” what follows a semicolon must be a full sentence.
“issue – the ‘mode collapse’ – appears” -> remove “the”
“complemented by a additional term” -> “an additional term”; I stopped tracking grammar mistakes after this point.
“MDGAN (Li et al., 2017),” -> MMD-GAN, MDGAN is (Che et al., 2017).
In table 1, does “time” column correspond to total training time of 25k iterations? Why is BuresGAN so much slower than MDGAN here but faster in iteration time according to appendix D table 16?

[Post rebuttal]
I've done a complete re-read of the updated manuscript. I am not convinced with the efficacy of the proposed method in solving mode drops beyond what has already been achieved in the literature. In particular, there is no guarantee that matching covariances within the feature space of a NN prevents mode dropping, as the NN (discriminator) itself must abstract away visual information to perform discrimination. The quality of writing has not improved significantly either. I am now more confident with my original assessment.

---

> ### Author Response · Authors · 2020-11-17
> **Initial comments and clarification**
>
> Thanks for your remarks.
>
> Since several reviewers suggested it, PacGAN will be added to our comparisons. Indeed, the PacGAN paper reports a perfect mode coverage in stacked MNIST experiment. The number of modes is computed thanks to a pretrained classifier which is not provided in PacGAN's repository, and therefore differs from the classifier that we use. The number of samples to evaluate as well as the architecture differ which makes a direct comparison impossible.
> We will add initial results for the PacGAN experiments before the end of the rebuttal.
>
> It is indeed instructive to add the FID scores for the resnet experiments since FID can also provide some information about generation quality. Our revised manuscript will contain FID scores for the resnet experiments.
>
> Concerning the training times, BuresGAN scales significantly better with respect to the feature map dimensions and data dimensions, although the training times are a bit longer for the lower dimensional synthetic examples. This is the reason behind the different ordering of the timings of the methods in Appendix D Table 16.
>
> The grammar of our manuscript will be rechecked. Any other suggestions are welcome.
>
> Additionally, if there are any other remarks we would be happy to discuss them in this forum.

---

### Official Review · AnonReviewer4 · 2020-10-26
**Interesting ideas but somewhat lacking in experimental evidence**

**Rating:** 6
**Confidence:** 4

**Review:**

The main strengths of the work are,
* The proposed idea is relatively novel, although similar ideas were explored in (Mroueh et al., 2017; Elfeki et al., 2019).
* The paper uses a computationally efficient expression (4) for the Bures distance. However, this expression has been proposed in prior work (Oh et al., 2020).
* The paper discusses connections with Wasserstein GAN and integral probability metrics. In particular, shows that the proposed distance os proportional to the 2-Wasserstein distance.
* Detailed information on the architectures and datasets is given in the Appendix, aiding reproducibility.
* The paper is very well written and easy to follow.

My main concerns are with the evaluation,
* Experiments on synthetic data: The following two papers outperform the proposed approach [1,2], especially in terms of number of high quality samples on both the Ring and Grid sets.
* Experiments on CIFAR-10 with DCGAN architecture - Both [1,2] again outperform the proposed approach in terms of the the FID metric. The order of the IvO score also does not seem to match prior works e.g. VEEGAN (Table 2), [1,2]. Please clarify the exact procedure used to compute the IvO scores.
* Experiments on CIFAR-10/STL-10 using ResNet architecture: The FID scores should be also reported for fair comparison with the state of the art. More importantly: the IS scores of best performing methods on CIFAR-10: ProgressiveGAN (Karras et al., 2017) and NCSN (Song & Ermon, 2019) are to reported for STL-10. It is unclear whether the proposed approach really achieves a new state of the art inception score on STL-10. Comparison with ProgressiveGAN (Karras et al., 2017) and NCSN (Song & Ermon, 2019) on STL-10 must be performed.
* The cost of the computing the Bures distance in terms of training time in comparison to simpler losses like Hinge loss [3] or gradient penalty loss of WGAN-GP should be clarified. It is unclear whether the additional resources required (if any) justly the limited performance gain of the proposed method.


[1] Mixture Density Generative Adversarial Networks, CVPR 2019.

[2] "Best-of-Many-Samples" Distribution Matching, NeurIPS Workshop, 2019.

[3] Spectral Normalization for Generative Adversarial Networks, ICLR 2019.

---

> ### Author Response · Authors · 2020-11-17
> **Initial comments and clarification**
>
> Thank you for your comments.
>
> Indeed, we did not compare with mixture density GANs. This is certainly of interest although we consider this modification as being 'orthogonal' to our approach since BuresGAN's objective can be also used in combination with mixture density GANs. Indeed,  Gaussian Mixture Models affect the generation, while our paper discusses the loss function.
> The aim of our comparison was to illustrate the improvement given by BuresGAN for a `fixed architecture in order to provide a fair comparison. The only changes between the methods given in our paper is possibly the choice of hyperparameters that we set to the values advised in the original papers.
>
> Though IS already indicates the generation quality, we agree that FID scores can be also instructive for assessing the high quality generations of CIFAR 10 and STL 10 data sets (Section 4) since it also relies on the data distribution. We are going to provide FID scores in the revised version of our manuscript.
>
> Concerning the cost of BuresGAN, its training is indeed slower for the synthetic examples compared to some other GANs used in the comparisons. However, BuresGAN scales better with the dimension of the input space and of the feature map. In the case of CIFAR-10 and STL-10, for example, BuresGAN is significantly faster compared to WGAN-GP.
>
> We are afraid that implementing and running Progressive GAN and NCSN on STL datasets ourselves is not feasible given the time constraints of the ICLR timeline.
>
> The IvO scores depend on the number of correction pairs and restarts, and the parameters of the optimizer. We emphasize that IvO scores cannot be compared between different papers if the same parameters are not used, although the comparison of the different scores within this manuscript is meaningful.
> If the reviewer is interested, the full code of the experiments, including the IvO evaluation code, is available in the supplementary material.
>
> Additionally, if there are any other remarks we would be happy to discuss them in this forum.

---

### Official Review · AnonReviewer1 · 2020-10-27
**Lack of experimental comparisons to the state-of-the-art methods on mode collapse**

**Rating:** 6
**Confidence:** 4

**Review:**

Update:

Thank the authors for providing the additional results and updating the paper. Since the comparison to one state-of-the-art method has been added (though some other state-of-the-arts are still missing) and the benefit is shown across different settings, I increase the score from 5 to 6.

(One suggestion: I would recommend you to highlight the changes in the revision with a different color, so that readers can identify the changes easily.)

----
----
The paper addresses the mode collapse issue in GANs. More specifically, the paper proposes to add a regularization which matches the Bures distance between the covariance matrices of the features of real and generated data. The paper demonstrates the performance across different datasets and architectures.

Overall, the paper has several merits:
* Experiments across a wide range of datasets and architectures.
* A detailed comparison of training time.

However, I cannot recommend this work for acceptance at this point, mainly because the paper did not compare with the state-of-the-art (and some widely-used) methods for fighting mode collapse, and the improvements on the benchmark datasets are rather weak. The details are below.
* The paper does compare with several GANs on the standard benchmark datasets for evaluating mode collapse (e.g. 2D-grid, 2D-ring, stacked MNIST). However, the baselines are rather weak and old. Even in that case, the scores of (Alt-)BuresGAN in Table 1 and Table 2 are just similar to (or sometimes even worse than) those baselines. In fact, there are many other newer and/or better methods that have shown to be outperforming the baselines the paper considered by a large margin, and some of them are already been out for years (e.g. [1,2,3,4]). But the paper did not compare with those methods. Therefore, it is unclear at all whether (Alt-)BuresGAN is useful or not.
* More specifically, on the stacked MNIST dataset, for example, many methods can dramatically improve the DCGAN baseline from ~100 modes (a relatively poor score) to 1000 modes (the best possible score on this dataset) [1,3,4]. The experiments in your paper seem to build on a better DCGAN architecture (993.3 modes already). But even in that case, (Alt-)BuresGAN only achieves 995.0 modes, and in fact, this score is within std to the DCGAN baseline. Also, on 2D-grid and 2D-ring datasets, some methods can achieve much better results and improvements than yours both quantitatively and qualitatively [2]. I understand that possibly the hyper-parameters and architectures in (Alt-)BuresGAN and those papers are different, so we cannot directly conclude that (Alt-)BuresGAN is worse than [1,2,3,4]. But these results do raise critical concerns about the performance of (Alt-)BuresGAN in fighting mode collapse, compared with the state-of-the-art methods.

Besides this point, I also have some other questions/suggestions:
* You use the features from the last layer of the discriminator. Why do you choose the last layer? How the performance would be if you are using other layers?
* "Algorithmic details" paragraph on page 4: the regularization term is already mentioned before, so you might consider removing it here.
* In the same paragraph, you might want to move "In the tables hereafter, we indicate the largest scores in bold if they differ from lower scores by at least one std" to the experimental section, because it is for result presentation, not for your algorithm.
* In fact, it is unclear to me if you are using this rule to mark the bold numbers. For example, in table 1, 84(6) and 22.9(4) shouldn't be marked as bold according to this rule. The same problem exists in all other tables in the paper. To me, how you mark the bold numbers seems random.

In conclusion, the paper does have some merits, but also have some critical problems, especially lacking the comparisons to the state-of-the-art methods, which makes it hard to judge the contribution of the method. **However, I am happy to adjust the scope if the authors can provide evidence regarding comparisons to the state-of-the-art methods during the rebuttal.**

[1] Lin, Zinan, et al. "Pacgan: The power of two samples in generative adversarial networks." Advances in neural information processing systems. 2018.

[2] Xiao, Chang, Peilin Zhong, and Changxi Zheng. "Bourgan: Generative networks with metric embeddings." Advances in Neural Information Processing Systems. 2018.

[3] Belghazi, Mohamed Ishmael, et al. "Mine: mutual information neural estimation." arXiv preprint arXiv:1801.04062 (2018).

[4] Eghbal-zadeh, Hamid, Werner Zellinger, and Gerhard Widmer. "Mixture density generative adversarial networks." Proceedings of the IEEE Conference on Computer Vision and Pattern Recognition. 2019.

---

> ### Author Response · Authors · 2020-11-18
> **Initial comments and clarification**
>
>  Thank you for your comments.
>
> Indeed, mixture density GANs are interesting. By nature, they are particularly suited to the mixture of Gaussians used in ring/grid examples. However, BuresGAN could also be considered in combination with mixture models, and therefore we view these two contributions as complementary. Additionally, we did not compare with many methods using a multiple generators and/or discriminator architecture since we simply propose a different objective function.
> For a fixed architecture, BuresGAN improves over the other cited methods, while it is also robust as we vary the networks' architectures.
>
> In order to have another more recent benchmark, we will add PacGAN to our comparisons, since a majority of the reviewers advised it. We will add some initial results for the PacGAN experiments before the end of the rebuttal.
> In comparison with our manuscript, in the PacGAN paper, a simpler DCGAN  architecture  gives worse results for the vanilla GAN. In our paper, we do not want to make the stacked MNIST experiment artificially harder by choosing a very simple architecture. **On the contrary, we re-use the same architecture as in VEEGAN, GDPP and other works that we compared with.**
>
> It is hard to compare with the setting of PacGAN paper, since the associated code repository does not provide the CNN classifier for counting the modes. Furthermore, in PacGAN paper, 26000 test points are generated to assess model collapse, while we use 10000 test samples in our manuscript. Therefore, it is not fair to compare the results reported in our paper with PacGAN paper. Indeed, for assessing the number of modes in stacked MNIST experiment, we claim that the relative performance comparison is meaningful given the same experimental settings, while cross-paper comparisons should be avoided. Finally, the packing approach can be also applied to any existing GAN, so that one could also consider a packed BuresGAN.
>
> We use the final layer because we assume that this final layer will give the most relevant features because that is the one used for the final classification.
> Concerning the bold numbers, thank you for spotting this mistake. We will make it more consistent
>
> Additionally, if there are any other remarks we would be happy to discuss them in this forum.

---

> > ### Comment · AnonReviewer1 · 2020-11-18
> > **Doubts**
> >
> > Thank you for the response!
> >
> > * **You said you "we re-use the same architecture as in VEEGAN, GDPP". I highly doubt if that's true.**
> >     * According to the VEEGAN paper (footnote 3), they are using the public implementation of DCGAN (https://github.com/carpedm20/DCGAN-tensorflow). But according to Table 8 in your paper, you are using a different architecture.
> >     * In VEEGAN paper, the number of recovered modes for the GAN baseline is 99. In your paper, your GAN baseline recovers over 990 modes. These are clearly coming from different architectures/hyperparameters.
> >     * I don't think you are using the same architecture as GDPP either. I checked both the code and architecture description in GDPP and your paper, and they are different (e.g. the number of convolution layers in the discriminator).
> >     * In GDPP paper, the number of recovered modes for the GAN baseline is 427. In your paper, your GAN baseline recovers over 990 modes. These are clearly coming from different architectures/hyperparameters.
> >
> > * **You said PacGAN uses a simpler DCGAN architecture and makes the stacked MNIST experiment artificially harder. HOWEVER,**
> >     * According to PacGAN's released code, PacGAN uses exactly the same architecture as VEEGAN, based on the public implementation of DCGAN (https://github.com/carpedm20/DCGAN-tensorflow).
> >     * According to the description in the papers, MINE and Mixture Density GANs also use the same architecture as PacGAN & VEEGAN.
> >     * Even in this hard setting as you said, PacGAN, MINE, and Mixture Density GANs can recover all 1000 modes, which demonstrates their performance.
> >
> > * From the above, it seems to me that you are making the stacked MNIST experiment artificially simpler, compared with what have been used in **many** prior papers.
> >
> > * **As I already mentioned in my initial review**, I totally agree that we cannot directly compare the results across papers and draw conclusions from them. However, since you didn't compare with many state-of-the-arts **(as both other reviewers and I pointed out)**, I have to use other ways to get a sense/guess of how Bures GAN performs. The point I was trying to say is that the improvement of Bures GAN on Gaussian mixture datasets and stacked MNIST datasets is relatively very small, even if the setting is simple as you agree.
> >
> > * Looking forward to seeing the results!
> >
> > Please free to comment/discuss/correct me if I miss something!

---

> > > ### Author Response · Authors · 2020-11-19
> > > **Clarification of parameters**
> > >
> > > Thank you pointing out a confusion aspect of the paper and for comparing the original implementations.
> > >
> > > For the stacked MNIST results, our implementation is based on the architecture given in the GDPP GAN paper. Only the learning rate of the optimizer (and $\beta_2$) and the latent distribution differ. For GDPP and BuresGAN, both the discriminator and generator have 3 convolutional layers. In the main part of our paper, we report these results in Table 2 for a discriminator with 3 conv layers.
> > >
> > > Table 18 in our appendix also gives results for a discriminator with only 2 convolutional layers cfr. the architecture described in Table 8. This might be the cause of the confusion. Our code allows to use any number of conv layers for the discriminator. We will add a Table in the appendix to also describe the architecture with 3 conv layers, in order to remove the ambiguity.
> > >
> > > The discriminator and generator of VEEGAN and PacGAN both have 4 convolutional layers and are therefore more complicated.
> > >
> > > Correction: it is then true that we do not have exactly the same architecture as VEEGAN and PacGAN as you correctly point out. Indeed, these differences in architecture and hyperparameters cause the difference in the number of recovered modes.
> > >
> > > To avoid confusion, the hyperparameters and architectures of the stacked MNIST experiment performed by the different methods are given below in a table, which represents our best efforts at providing a comparison between the different parameters used.
> > >
> > > | Parameters           | BuresGAN               | PacGAN                 | VEEGAN                 |GDPP                              |
> > > |----------------------|:------------------------:|:------------------------:|:------------------------:|:-----------------------------------:|
> > > | learning rates       | 1 x 10^(-3)            | 2 x 10^(-4)            | 2 x 10^(-4)            | 1 x 10^(-4)  **      |
> > > | learning rate decay  | no                     | no                     | no                     | no **                |
> > > | Adam beta_1          | 0.5                    | 0.5                    | 0.5                    | 0.5                               |
> > > | Adam beta_2          | 0.999                  | 0.999                  | 0.999                  | 0.9                               |
> > > | iterations           | 25000                  | 20000                  | ?                      | 15000*                            |
> > > | disc. conv. layers   | 2 / 3                  | 4                      | 4                      | 3                                 |
> > > | gen. conv. layers    | 2 / 3                  | 4                      | 4                      | 3                                 |
> > > | z dim.               | 100                    | 100                    | 100                    | 128                               |
> > > | batch size           | 64 / 128 / 256         | 64                     | 64                     | 64                                |
> > > | evaluation samples   | 10000                  | 26000                  | 26000                  | 26000                             |
> > > | z dist.              | normal                 | uniform[-1,1]          | normal                 | uniform[-1,1]                     |
> > >
> > >
> > > Hyperparameters for the Stacked MNIST experiments as reported in the corresponding papers and code repositories. The GDPP paper used 30000 iterations for training DCGAN and unrolled GAN (indicated by *). **  means as found on Github.
> > >
> > > We agree that the differences in Table 2 for stacked MNIST are not always significant. Many methods can retrieve almost all modes on stacked MNIST.

---

> > > > ### Comment · AnonReviewer1 · 2020-11-19
> > > > **Thank you for the clarifications and corrections!**
> > > >
> > > > Thank you for the detailed clarifications and corrections! (Just one more minor correction: I think VEEGAN is using uniform[-1,1], if you look at the history of the public DCGAN repo at the time the paper was published).
> > > >
> > > > I guess the conclusion of this discussion is that on stacked MNIST dataset, you are using different architectures and/or hyperparameters compared with prior work, which makes it easier for the GAN baseline. This is fine, again, as long as you can provide experimental comparisons to the state-of-the-arts (which you said you will add). Looking forward to the results :)

---

> > > > > ### Comment · AnonReviewer1 · 2020-11-24
> > > > > **Thanks for the revision.**
> > > > >
> > > > > Thank the authors for providing the additional results and updating the paper. Since the comparison to one state-of-the-art method has been added (though some other state-of-the-arts are still missing) and the benefit is shown across different settings, I increase the score from 5 to 6.
> > > > >
> > > > > (One suggestion: I would recommend you to highlight the changes in the revision with a different color, so that readers can identify the changes easily.)

---

### Official Review · AnonReviewer3 · 2020-10-28
**The experiments should be designed and evaluated thoroughly**

**Rating:** 5
**Confidence:** 4

**Review:**

*Update after reading the authors' rebuttal:
The revision of the manuscript was much improved. However, the lack of controlled experiments does not convince me. This paper proposes a new penalty to deal with mode collapse, and the authors claimed that it could be easily added to any existing GAN variants. The authors may need to provide a controlled experiment to show their claim, e.g., what happens if adding their penalty to some GAN variants, fixing the same setting.
One good point from the new revision is that BuresGAN can be competitive with the state-of-the-art baselines, with some slight changes in the network architectures. I suggest the authors to test their penalty with other GAN variants to stronger support their claim.

--------------------------------------------
This paper concerns the problem of mode collapse in Generative Adversarial Networks (GANs). A new measure (Bures distance) is investigated to overcome mode collpase in GANs. Bures distance can help us to measure the similarity of two possitive semi-definite matrices and was proposed before. This paper proposes a penalty to the generator loss to encourage the diversity of fake data to match the diversity of real data. To do this, the last layer (providing the representation for each input) of the discriminator is used to define the diversity in input. Bures distance is then used to define the similarity between the diversity of real data with that of fake data, leading to the novel method called BuresGAN. Four datasets are used for their evaluation and comparison with 7 baselines. The experimental results seem to be promising.

Pros:
- Bures distance is an interesting metric and promising to deal with mode collapse.
- The experimental results are promising.

Coins:
- Unclear context: mode collapse is a challenging problem. There exist various approaches to deal with this problem, such as using more generators, more discriminators, using different losses, or penalty. However, this paper does not provide an extensive overview of the existing literature on mode collapse. As a result, it is unclear about the context of this paper and the significance of their contributions. The authors should provide an extensive summary and then place their work in a clear context.
- Unclear significance: The authors use different network architectures for different methods in their experiments, e.g., MDGAN often uses architectures which are different from other methods. Such different architectures make us hard to see which components (e.g. architecture, loss, penalty) really contribute to the success/failure of a method. Also, it is unclear whether the good performance of BuresGAN comes from the new penalty or not. The authors should design a controlled experiment to see the practical effect of their new penalty compared with other penalties, such as by fixing the shared architectures for generator and discriminator and their losses.
- Baselines: some other types of baselines for dealing mode collapse should be included, e.g. multi-generator or multi-discriminator based methods. Such a comprison will provide more evidents to see the significance of the proposed penalty. Some examples are MAD-GAN [Ghosh et al., 2018] and D2GAN [Nguyen et al., 2017].

Minor comment:
- The results of some baselines, e.g., MDGAN, VEEGAN, UnrolledGAN, are sometimes not very good as reported in Table 3. Why this happened? Was it because of the use of default settings or unconvergence when training?

Reference:
- Nguyen, T., Le, T., Vu, H., & Phung, D. (2017). Dual discriminator generative adversarial nets. In Advances in Neural Information Processing Systems (pp. 2670-2680).
- Ghosh, A., Kulharia, V., Namboodiri, V. P., Torr, P. H., & Dokania, P. K. (2018). Multi-agent diverse generative adversarial networks. In Proceedings of the IEEE conference on computer vision and pattern recognition (pp. 8513-8521).

---

> ### Author Response · Authors · 2020-11-17
> **Initial comments and clarification**
>
> Thank you for your remarks.
>
> Concerning the presentation of the related work, we will try to improve the description of the context in the revised version of our manuscript.
>
> GANs with multiple generators and/or discriminators are certainly of interest. We did not compare with these GANs since their approach is complementary to ours.
> To introduce another recent benchmark, we will include PacGAN since it has been advised by several reviewers.
> We will add some initial PacGAN experiment results before the end of the rebuttal.
>
> It can be noticed from PacGAN paper, that VEEGAN and DCGAN also have very bad performance in some cases. This might be due a sensitivity to the architecture choice. MDGAN and VEEGAN often have problems with non-convergence for the given architecture, and many other architectures that we have tested. It should be mentioned that there were runs where these methods achieved very good results but they were inconsistent, as indicated by the standard deviation.
> The results in our manuscript were obtained for the parameters proposed in the original papers, while the architecture is the same for all the methods.
>
> Additionally, if there are any other remarks we would be happy to discuss them in this forum.

---

### Author Response · Authors · 2020-11-24
**Overview of changes in the new version of the paper**

The paper has been updated according to the feedback received by the reviewers.
An overview of the changes can be found below.

Major changes:

- High quality generation (CIFAR-10 and STL-10): we added the FID scores to our results for the resnet experiments. For the STL-10 dataset, we noticed that the scores reported by the other methods had been obtained for a rescaled size of 48x48x3. Therefore, we ran our simulations again for this setting and modified the corresponding entries in the table. To the best of our knowledge, the FID scores obtained by BuresGAN improve over the results available in the literature for a Resnet architecture on STL-10.

- In order to compare with a more recent method advised by most of the reviewers, we repeated our experiments for evaluating mode collapse with PacGAN. The upshot is that PacGAN indeed achieves good results although BuresGAN is often competitive or yields better results. The tables were modified accordingly.

- There was no significant winner in the original stacked MNIST experiment. Therefore a similar experiment, following the PacGAN and VEEGAN paper, with a more challenging architecture which includes 4 convolutional layers for both the generator and discriminator is included. BuresGAN outperforms PACGAN and other direct competitors, only  WGAN-GP is capable of getting a better result. However note that our empirical results indicate that WGAN-GP is sensitive to the choice of architecture and hyperparameters while its training time is significantly longer. This experiment further confirms that BuresGAN is very consistent in performance over different architectures.

Minor changes:

- The grammar was rechecked and several sentences were rephrased to improve the readability.

- To avoid any confusion, in all tables, bold values now indicate the best scores.


Thanks again to all the reviewers for their feedback.

---

### Decision · Program_Chairs · 2021-01-07
**Final Decision**

**Decision:**

Reject

**Comment:**

The authors propose the Bures metric (a distance between covariance matrices of the last feature layer of a discriminator) as an extra loss to mitigate mode collapse. The metrics bears some similarity to the covariance term in FID, and builds upon a number of GAN papers that augment GAN losses with differences in covariances between real and generated data. As the reviewers noted, the authors did an admirable job of performing an apples-to-apples comparison with other GAN alternatives, and use a number of metrics to demonstrate their results. Unfortunately, the most extensive comparisons usedthe DCGAN architecture, which is now 2-3 years too old for a potential reader to ascertain how well the proposed method would work on her problem. Moreover, the reviewers identified discrepancies in the baselines of those the experiments, as the numbers reported in this paper seemed to indicate poorer performance and the numbers reported in the original papers.

During the rebuttal phase, the authors demonstrated that these methods also perform well with using ResNet architectures on CIFAR-10 and STL-10, and the method is competitive with more recent models. As noted by the reviewers, however, these new comparisons are not as extensive and controlled as those that used DCGAN. Furthermore, results on more difficult datasets, such as ImageNet, are missing.

Had the extensive experiments used ResNets instead of DCGAN, or if the authors demonstrated promising results on ImageNet, I would recommend acceptance. Unfortunately, I think the audience for this paper in 2020-2021 would be relatively limited, so I have to recommend rejection.